# Aquaphotomic Study of Effects of Different Mixing Waters on the Properties of Cement Mortar

**DOI:** 10.3390/molecules27227885

**Published:** 2022-11-15

**Authors:** Jelena Muncan, Satoshi Tamura, Yuri Nakamura, Mizuki Takigawa, Hisao Tsunokake, Roumiana Tsenkova

**Affiliations:** 1Aquaphotomics Research Department, Graduate School of Agricultural Science, Kobe University, Kobe 657-8501, Japan; 2Technical Department, ISOL Technica Corporation, Kyoto 606-0022, Japan; 3Institute of Engineering, Graduate School of Engineering, Division of Urban Engineering, Osaka Metropolitan University, Osaka 599-8531, Japan

**Keywords:** cement concrete, mortar, water, water molecular structure, drying shrinkage, aquaphotomics, near infrared spectroscopy

## Abstract

The mixing water used for cement concrete has a significant effect on the physical properties of the material after hardening; however, other than the upper limit for the mixed impurities, not enough consideration has been given to the functions and characteristics of water at the molecular level. In this study, we investigated the effect of four different types of water (two spring-, mineral waters, tap water and distilled water) on the drying shrinkage of the hardened cement by comparing the material properties of the concrete specimens and analyzing the molecular structure of the water and cement mortar using aquaphotomics. The near infrared (NIR) spectra of waters used for mixing were acquired in the transmittance mode using a high-precision, high-accuracy benchtop spectrometer in the range of 400–2500 nm, with the 0.5 nm step. The NIR spectra of cement paste and mortar were measured in 6.2 nm increments in the wavelength range of 950 nm to 1650 nm using a portable spectrometer. The measurements of cement paste and mortar were performed on Day 0 (immediately after mixing, cement paste), 1 day, 3 days, 7 days, and 28 days after mixing (cement mortar). The spectral data were analyzed according to the aquaphotomics’ multivariate analysis protocol, which involved exploration of raw and preprocessed spectra, exploratory analysis, discriminating analysis and aquagrams. The results of the aquaphotomics’ analysis were interpreted together with the results of thermal and drying shrinkage measurements. Together, the findings clearly demonstrated that the thermal and drying shrinkage properties of the hardened cement material differed depending on the water used. Better mechanical properties were found to be a result of using mineral waters for cement mixing despite minute differences in the chemical content. In addition, the aquaphotomic characterization of the molecular structure of waters and cement mortar during the initial hydration reaction demonstrated the possibility to predict the characteristics of hardened cement at a very early stage. This provided the rationale to propose a novel evaluation method based on aquaphotomics for non-invasive evaluation and monitoring of cement mortar.

## 1. Introduction

Cement concrete is composed of cement, water, fine aggregate (sand), coarse aggregate (gravel), and admixture. The hydration reaction of cement starts immediately when it comes into the contact with water, and cement hydrates are formed as the cement gradually hardens. Generally, the material properties of cement concrete and the quality of cement depend on the weight ratio of water to cement (water–cement ratio, *w/c* ratio) [1,2,3,4,5]. This number *w/c* is in inverse correlation with the concrete strength: the smaller the *w/c* ratio, the greater the strength, durability and watertightness [3]. The *w/c* ratio is linked to the spacing between the cement particles in the cement paste—if the spacing is smaller, the process of filling in the gaps between cement particles by cement hydrates is faster and the links created by the hydrates are stronger, hence the stronger the concrete [3]. Another important aspect related to the water-to-binder (or water-to-cement) ratio is the microstructure, and especially the nanoscale characteristics [6]. For example, the dense microstructure provides the excellent mechanical properties and long-term service performance to ultra-high performance concrete (UHPC), an advanced cement-based [7,8]. The UHPC mixtures usually have a low water-to-binder ratio, and a higher content of cement and silica fume particles [8], which, as research studies showed, mainly play the filling role, and the hydration degree of cement is only around 30–35% [9]. On the other hand, a high water-to-binder ration leads to high risks for cracking due to autogenous shrinkage, which is closely related to mechanical properties and durability [6]. The incorporation of various organic or inorganic modifiers in cement is another method of effectively influencing the water-to-binder ratio due to the modified binder properties, which can lead to improved volume stability and water resistance [10]. Therefore, it is important to control the water–cement ratio in order to provide resistance to neutralization (carbonation-reaction with the carbon dioxide from the atmosphere) and infiltration of salt (chloride damage) into cement concrete—the two main causes of chemo-mechanical changes and deterioration of the durability of the concrete structure [5].

The shrinkage of cement concrete is a phenomenon that can occur in any concrete structure, and cracks and dimensional changes due to shrinkage can greatly affect the state of stress in the structure and various performance parameters, including durability. The cement shrinkage is one of the classical research subjects in concrete engineering; the elucidation of the mechanism of shrinkage, theoretical and various other models and prediction methods have been proposed [11,12,13,14,15,16,17,18,19,20]. The shrinkage behavior is associated rather with the microscale thermodynamic properties such as hydration, pore-structure formation, and the water status in micropores [20], than with the macroscale properties such as the *w/c* ratio [21]. This is because the shrinkage behavior cannot always be evaluated only by the *w/c* ratio. In order to explain the microscale aspects of the drying shrinkage principle, several theories concerning the water status in micropores have been proposed to explain the mechanism that causes drying shrinkage: capillary tension theory (caused by the water meniscus), disjoining pressure theory (caused by water films in narrow pores), surface tension theory (surface energy change caused by water desorption), and interlayer water transfer theory [20,21,22,23,24,25,26,27,28].

Since the size of the micropores in cement matrix varies greatly, the water status can be quite different, and the theories explaining the shrinkage behavior are usually combined to provide an explanation for the behavior at various levels of relative humidity (RH) [20]. Capillary tension and disjoining pressure are considered to be the predominant mechanisms in the mid to high humidity range, and surface tension and interlayer water transfer in the low humidity range [20]. However, there is currently no unified theory that can explain the shrinkage behavior of hardened cement over the entire humidity range. Since water, fine aggregate, and coarse aggregate in cement concrete are generally collected and procured locally at the place where cement concrete is manufactured, the physical properties of the materials, composition, and curing conditions (environmental conditions) can vary greatly. This makes it additionally difficult, considering the great many number of variables, to efficiently investigate the microscopic mechanisms causing the shrinkage, and to understand the shrinkage behavior of hardened cement. Therefore, there is a great need for the establishment of a universal and comprehensive analysis method that can take into consideration all these aspects.

Given the background explained above, and especially the role of water in the shrinkage process, it is evidently necessary to examine various chemical and physical properties of water because they will directly influence the physical properties of cement concrete after hardening. In regards to the conditions that water should satisfy in order to be used for the production of cement concrete, the standards such as JIS A 5308—Ready-Mixed Concrete Appendix C [29] and ASTM Designation C94-96—Standard Specification for Ready-Mixed Concrete [30], only specify that the water used should be clean potable water. However, even the sludge and well water are sometimes used, and there are no specific requirements other than the upper limit of harmful substances such as impurities and chloride ions [30,31].

In recent years, various types of “functional water” have been developed, that is, waters whose functionality is supposedly enhanced to serve a specific purpose for health and well-being [32,33], or other purposes in medicine and agriculture [34,35,36,37,38,39]. In the science of cement materials, studies have demonstrated that the use of “functional water”, for example magnetically treated water or hydrogen nano-bubble water, can effectively improve the compressive strength, workability, and watertightness of cement concrete, even reducing the needed amount of cement in the mix [40,41,42,43,44,45,46,47,48]. However, since the mechanism of action of “functional water” on the physical properties of the hardened cement has not been investigated at the molecular level, the effectiveness of “functional water” cannot be quantitatively evaluated for full-scale practical use and warrants further investigations [48,49].

The methods used to date to investigate the cement hydration include isothermal calorimetry, thermal analyses, monitoring of chemical shrinkage, in situ quantitative X-ray diffraction, nuclear magnetic resonance spectroscopy (NMR), quasi-elastic neutron scattering (QENS), and small angle neutron scattering (SANS). They have proven useful for comparing the hydration of different cements, and especially in combination, they can provide insights into the hydration process that cannot be obtained by any one method alone. However, all of them monitor the overall progress of hydration, without providing more insight into the nature of the occurring chemical reactions or resolving the details of particular underlaying mechanisms [50]. In particular, they cannot be exploited on a massive scale in practical applications.

In this study, aquaphotomic near infrared (NIR) spectroscopy [51] is proposed as a suitable method for the characterization of both mixing water and cement mortar. This method is based on the utilization of the light–water interaction, which provides the information about the state of the water molecular network within the material, and indirectly about the material itself by analysis of the NIR spectra [52]. The aquaphotomics’ science and technology have been gaining worldwide attention in recent years due to the noninvasiveness, ease of use, wide field of applications, and novel discoveries [52,53,54]. Further, considerable efforts have been invested in the developments of spectral preprocessing [55] and chemometrics techniques [56,57,58,59,60,61] for analysis of NIR spectra that contributed to the better understanding of the water molecular structure. Aquaphotomics has been providing a basic understanding of the water molecular network-related functionalities and new technical solutions for water and food quality monitoring, biometrics, biological diagnosis, biological monitoring, and water quality monitoring in waterworks facilities, but construction materials such as hardened cement have not been explored enough [52].

Therefore, in this study, the effects of four different types of water on the physical properties of the hardened cement were investigated. For the purpose of this work, the focus will be placed only on the drying shrinkage characteristics of the hardened cement, while other properties such as the compressive and bending strength, elastic modulus and others will be the object of further investigations. The results of this research will demonstrate that using the same cement components, but different waters for mixing, despite their negligible differences in mineral components, leads to different drying shrinkage properties of the produced cement. Further, using aquaphotomics for the characterization of waters, it will be demonstrated that the different water molecular structure is the reason for this because it dictates the different formation of cement matrix in the paste and mortar. The entire process can be monitored immediately upon the mixing of cement, which is used to predict the cement hardness at a very early stage and also monitored throughout the course of the cement hardening.

## 2. Results and Discussion

### 2.1. Mineral Content of Waters Used for Preparation of Cement

The results of the Inductively Coupled Plasma Mass Spectrometry (ICP-MS) analysis are presented in Table 1 for all the water samples used as mixing waters in preparation of the cement. Distilled water (W_dist_) was water from which minerals were removed, although trace amounts of Ca and Mg were still detected. The other three types of water had a higher mineral content compared to W_dist_. Mineral waters from the shallow source (W_shallow_) and mineral water obtained by mixing two mineral waters from shallow and deep source (W_mix_) had higher contents of Na, Si, and Ca compared to the Osaka city tap (W_tap_) water.

Minerals that have a relatively high content are highlighted in the Table 1 and the comparison of their amount in waters used for cement preparation is shown in Figure 1. The Si content is particularly high in mineral waters compared to W_tap_ and W_dist_. In addition, the content of Na, Mg, and Ca in W_mix_ is more than twice the amount in W_tap_ and W_shallow_. Even though there are large differences in the content of the five minerals (Na, Mg, Si, K, and Ca) (Figure 1), this is not considered relevant in the usual practice, and all four waters satisfy the tolerable limits of constituents according to the standards for most countries [62].

To conclude, according to the contemporary standards for mixing water, each of the four waters can readily be used for preparation of cement concrete. The existence of differences in constituents would not be considered relevant or of any consequence for the properties of concrete.

### 2.2. Aquaphotomic Characterization of Mixing Water

The characterization of mixing waters was performed using aquaphotomics’ NIR spectroscopy. The NIR spectra of mixing waters, acquired in the range 400–2500 nm, at a controlled temperature of 25 °C were trimmed to a region of 1300–1600 nm that corresponds to the first overtone of water stretching vibrations, which typically has a maximum around 1450 nm (Figure 2). The NIR spectra of waters are broad and the differences between the individual spectra are very small. In order to extract and emphasize the differences between the mixing waters’ spectra, the use of multivariate data analysis is required.

Therefore, the spectra were analyzed according to the protocol of the aquaphotomic spectral analysis [63]. Difference spectra, Principal Component Analysis (PCA) [64], Soft Modeling of Class Analogies (SIMCA) [65], and Partial Least Squares Regression (PLSR) Analysis (using temperature and consecutive irradiation as dependent variables) were performed (results not presented) in order to find the representative water absorbance bands—water matrix coordinates (WAMACS) [51]. The WAMACS are then used to depict the characteristic water spectral patterns (WASPs) of mixing waters on aquagrams [63,66] related to the respective functionalities. The selection of WAMACS was performed as described in the recent literature [63,67], by choosing the most consistently repeating and most influential absorbance bands in the entire performed analysis.

The calculated aquagrams of four mixing waters are presented in Figure 3, with 15 radial axes defined by the chosen WAMACS, displaying normalized absorbance values after the correction of spectra by standard normal variate (SNV) transformation [68] to cancel the potential baseline effects. The aquagrams of mixing waters demonstrate succinctly their water spectral patterns, compared to the ultra-pure water (labeled as EC, zero line on the graph, Figure 3).

Each radial axis on the aquagram, i.e., wavelength corresponds to the absorption of particular water molecular species. The resulting aquagrams present the water absorbance spectral patterns of each examined water and effectively convey the differences in their molecular structure. The wavelengths presented can be attributed to the absorbance bands of the following water species: 1347 nm represents OH group asymmetric stretching vibration ν_3_ and is the common band for all proton hydrates, 1360 nm is water solvation shell OH-(H_2_O)_n_ where n = 1, 2, or 4; 1375 nm is attributed to combination of symmetric and asymmetric stretching vibration ν_1_ + ν_3_; 1380 nm is water solvation shell OH-(H_2_O)_n_ where n = 1, 4 or superoxide tetrahydrate O_2_-(H_2_O)_4_; 1397 nm is water confined in the local field of ions; 1410 nm corresponds to free water molecules (S_0_); 1428 nm is hydration water–hydroxide and OH-(H_2_O)_4_ water shell; 1436 nm and 1444 nm are absorbance bands of protonated water dimer (Zundel Cation) and water dimers (S_1_); bands 1448 nm and 1453 nm can be assigned to solvation shells OH-(H_2_O)_n_, where n = 4, 5; 1460 nm to water molecules with two hydrogen bonds (S_2_), 1472 and 1480 nm to water molecules with three hydrogen bonds (S_3_), 1492 nm to water molecules with all four hydrogen bonds (S_4_); 1510 nm to the combination of symmetric stretching and bending vibrations ν_1_ + ν_2_; and strongly bound water [51,69,70,71,72,73].

In simple words, the bands located at the right side of the aquagram (1347–1444 nm) can be said to encompass the absorbance bands of free water, quasi-free water, weakly hydrogen-bonded water, and water involved in the hydration of solutes (solvation shells), while the left side of the aquagram (1444 nm to 1510 nm) represents absorbance bands of hydrogen-bonded water. Compared to the pure water, the spectral patterns of all mixing waters demonstrate higher absorbance at the right side of the aquagram, showing much higher ability for solvation. The water molecular structure of these waters resembles the ones that could be observed at increased temperatures. The largest differences in spectral patterns between the four types of waters can be found in the region of 1428 to 1460 nm, in particular at 1448 nm—bands that can be attributed to the absorption of water molecules in solvation shells, which feature four or five water molecules.

These results demonstrate that, regarding the composition, despite what seems to be negligible amounts of ions, the mixing waters do not have the same molecular structure, and they are quite different compared to the ultra-pure water, and furthermore, even the W_dist_ shows a distinctive spectral pattern. It could be said that, compared to the spectral pattern of pure water at 25 °C, all the examined waters demonstrate the molecular structure that would correspond to the molecular structure of pure water at higher temperatures [74]. This especially can be of significance with respect to the solvation ability.

### 2.3. Aquaphotomic Characterization of Cement Mortar

#### 2.3.1. Raw and Transformed near Infrared Spectra of Cement Mortar

Raw NIR spectra acquired for the mortar on the day it is prepared (cement paste, Day 0) and during the process of setting and hardening (cement mortar, after 1, 3, and 7 days) are presented in Figure 4a.

The absorbance spectra demonstrate large baseline effects due to the influence of scatter and are mainly overlapped, making it difficult to observe differences depending on the water used in the mixing of the concrete. However, the broad band centered around 1450 nm, corresponding to the first overtone of water stretching vibrations, can still be observed.

To emphasize the chemical information in the spectra, the transformation using the linear detrend correction and SNV transformation [68] was performed and the spectra after correction are presented in Figure 4b. The pre-processing eliminated the baseline differences efficiently and at three places in the spectra interesting features appeared—at 1360 nm, 1410 nm, and 1460 nm. At these specific places, some differences in the spectra of mortar could be observed even with the naked eye. All three bands are, as mentioned before, well-known water absorbance bands that can be attributed to solvation shells, free water molecules, and water molecules with two hydrogen bonds.

This suggests the importance of these water molecular structures for the description of the process of change in cement mortar over time, and as a function of the molecular structure of water used for mixing. To explore these differences further, the multivariate analysis tools were employed.

#### 2.3.2. Principal Component Analysis (PCA) of Cement Mortar

The exploratory analysis in the form of PCA [75] was applied on the spectral data separated in four datasets according to the type of water used for mixing. The separation was performed with the aim to take the full advantage of pre-processing using detrend and SNV and correct the baseline effects appropriately, since the origin of scatter is physical in nature, and therefore may vary between the different mortars. The smoothing using Savitzky-Golay 2nd order polynomial filter [76] and 21-point window size was also performed to eliminate the noise from the spectra.

The majority of variance (more than 95%) in each dataset was captured by the first two principle components PC1 and PC2. The PC1-PC2 scores plots of PCA analyses are presented in Figure 5, for mortar created using W_dist_ (Figure 5a), W_tap_ (Figure 5b), W_shallow_ (Figure 5c) and W_mix_ (Figure 5d), where the scores are colored depending on the day of the spectral acquisition. In all four cases, the first two principal components explained more than 95% of variations in the datasets (Table 2) and the patterns of scores corresponding to the different days (the day of cement paste preparation—Day 0 and cement mortar after 1, 3, and 7 days) could be well-distinguished in the PC1-PC2 spaces.

While so far in the analysis the minute quantities in ions and their differences between mixing waters might have seem negligible, and even the water spectral patterns similar, from the PC1-PC2 score plots it now becomes evident that the use of different waters for the mixing of cement results in quite different characteristics during the cement setting. First, it can be noticed that the spreading of the scores differs, and it is largest on Day 0 in cement paste, immediately after the mixing, suggesting large differences across the cement paste. However, after the mortar is set and left to harden, the differences diminish over time, suggesting that the water in cement paste is still in a variety of states, compared to the subsequent days. The scores corresponding to the Day 0 are for all water types located in the negative part of PC1 and mostly positive part of PC2, with the exception of W_shallow_, whose scores are located neutrally along the zero line of PC2. This clearly indicates that cement paste created with W_shallow_ is very different compared to others, while the smaller spreading of the scores may indicate more uniform characteristics of cement paste.

The pattern of changes along time, as the cement is aging, can be observed mostly along the PC1. This means that the PC1 describes the transformation of water in the cement during this process. Especially in regards to the time trend, the differences increase. The scores corresponding to the spectra acquired after 1 day are, in the case of all waters, located in the negative parts of both PC1 and PC2, except again in the case of W_shallow_. It seems that in the case of W_shallow_, the cement mortar is not changed much during first 24 h, as can be observed from Figure 5c—the scores of Day 0 almost coincide with the scores after 1 day.

The scores corresponding to the age of 3 and 7 days in all four cases are located in the positive part of PC1, but there are particular differences among the waters. The most well-defined groups of scores for 3- and 7-days age can be observed for W_tap_ (Figure 5b), suggesting that changes still take place in this cement. On the contrary, in the scores of W_dist_ (Figure 5a), the groups of scores corresponding to the 3rd and 7th day coincide, indicating very small changes after 3 days. The scores corresponding to these days of cement age in the cases of W_shallow_ and W_mix_ water partially overlap; however, the time trend is the opposite compared to the W_tap_, where the scores corresponding to the 7 days are located more closely to the Day 0 scores. From these observations, it can be concluded that the waters used for mixing cement mortar influence the uniformity of the mixture (cement paste, Day 0) and the behavior during the aging of the cement mortar during the first 7 days. The time trend of the changes is mostly described by first two principal components, but the changes during the first 24 h are mainly described by the PC2. The loadings of PC1 and PC2 are presented in Figure 6 for all four datasets.

From the loadings’ plots, the first thing to observe is the very similar shape of PC1 loadings for all four examined cases, and the occurrence of almost the same bands: the negative peaks at 1391 nm and 1397 nm, and positive peaks at 1472 and 1478 nm. The first two may be attributed to the, so-called, trapped water while the latter ones to the water molecules with three hydrogen bonds. Looking at the position of scores at the scores’ plots (Figure 5) and relating it to the sign of the observed peaks, it can be concluded that from the Day 0 during the first 24 h there is an increase in trapped water, but from that day onwards, the water in cement after 3 and 7 days is characterized by increased amount of water that is hydrogen–bonded. This may also account for the loss in the spreading of the scores on the 3rd and the 7th day, because the hydrogen bonded water is not easily changed.

The differences in bands 1391 nm and 1397 nm may be in the nature of ions or the confinement which entrap single water molecules, while the differences in 1472 and 1478 nm may be due to the isomerism of water molecular species with three hydrogen bonds. In the study of wood stiffness and strength properties, by Fujimoto, Yamamoto and Tsuchikawa 2007, the band located at 1476 nm was related to the semi-crystalline regions in cellulose [77]. The bands in our work (1472 nm, 1478 nm) may also indicate that the state of water in the cement matrix is semi-crystalline as the cement ages to the 3rd and 7th day. The confined water observed at 1391 nm or 1397 nm can also be understood as an interlayer between the sheets of calcium hydroxide Ca(OH)_2_, similar to the findings of Kondo et al. 2021 for the hydration and dehydration of Mg(OH)_2_ [78]. In any case, this entire process can be described as a transformation of weakly hydrogen bonded water to hydrogen bonded water, and as such, to use analogy with temperature it could be compared to the cooling of water within the cement matrix, which is consistent with the release of heat during the reaction of cement hydration [50].

On the other hand, the loadings of PC2, which is more related to the changes during the first 24 h, and therefore to the initial reaction of cement hydration, demonstrate somewhat different shapes depending on the water used in mixing. The common features are positive peaks at 1360 and 1366 nm, both of which correspond to water solvation shells, and a positive peak at 1472 nm, and again the water molecules bonded with three hydrogen bonds. Here, similarly, the difference in the position of the bands of solvation shells might be due to the differences in ions present in the waters, and their solvation. Or more likely, it is due to the different coordination number; Kondo et al. 2021 have reported band 1362 nm to correspond to OH-coordinated with 1 or 2 Mg^2+^ on the corner and edge of the Mg(OH)_2_ surface, while 1368 to OH-coordinated with 3 Mg^2+^. Having this in mind, the bands observed in PC2 loadings could be related to the Ca^2+^ ions. It is interesting to notice that there is a report of the absorbance band at 1366 nm, as related to the absorbance of a compound highly correlated with hardness [79]. This report was about the hardness of wheat, modeled based on the water extracts of whole wheat flours. Using selected treatments, it was established that the active compound was not a protein, lipid, hemicellulose, nor sugar. Based on the findings of this report, it may be concluded the band 1366 nm is very likely a water absorbance band related in general to hardness. Since the entire process of cement curing is about the loss of moisture and consequent hardening of cement, the observed absorbance band can be related to the hardness of cement mortar.

The negative peak common for loadings of PC2 for W_dist_, W_tap_ and W_mix_ appears at 1428 nm, while in the case of W_shallow_ at 1441 nm. The first one is assigned to hydroxide and hydration water, while the second one to the water dimer [51]. The importance of band 1428 nm, found to be related to amorphous regions in cellulose matrix of the wood and its stiffness and strength [77], may indicate not only the different state of water, but also the macro-scale properties of cement mortars made using W_tap_, W_mix_ and W_dist_. Particular differences occur at the region of hydrogen bonded water, where positive peaks and/or shoulders could be observed at 1503 nm in the case of W_tap_ and W_mix_, at 1492 and 1510 nm for W_dist_ and 1515 nm for W_shallow_. Looking again, together at the position of scores and sign of the peaks on loading plots, the process of cement hydration during the first 24 h can be described as a transformation of hydrogen-bonded water (>1492 nm) and water participating in ion solvation (1360 nm) to, in most cases, hydration water (1428 nm). The W_shallow_ should be excluded from this explanation, as the PC2 in this case is not related to the changes in the aging of cement; it only describes some variations within the groups of scores for each of the days. It is interesting to note, however, that water dimers were the water species connected to preservation of biological structures during the complete desiccation of resurrection plants, and were related to the glassy state of water due to the sugar–water interaction [80]. This finding may indicate the existence of that type of glassy structure of water in the mortar made by W_shallow_.

#### 2.3.3. Soft Modeling of Class Analogies (SIMCA) of Cement Mortar

SIMCA analysis was performed on datasets of cement paste and mortar spectral data and separated according to the type of water used. Each dataset, after separation, was first smoothed using the Savitzky–Golay 2nd order polynomial filter (21 points) and baseline corrected using linear fit and SNV transformation, considering that each spectral dataset can have specific baseline variations. Following the pre-processing, each dataset was subjected to SIMCA analysis in order to discriminate between different days when the spectra were acquired, specifically on the day when the cement paste was first prepared (Day 0), and after leaving the mortar to cure (After 1 day, 3 days, and 7 days).

The results of the analysis, first presented in Table 3, demonstrate that irrespectively of the water used for cement preparation, the discrimination accuracy between the spectra of cement mortar acquired on different days was higher than 90%, and the interclass distances were larger than three, demonstrating very good, reliable separation [81,82] and indicating large changes in cement over time. However, the trends in the values of interclass distances were quite different depending on the water used in mixing, especially for the 3rd and 7th day, which implies differences in the hardening of mortar over time.

To investigate which variables, i.e., wavelengths in the spectra carried most information with regards to their discrimination with respect to different days, the discriminating powers of SIMCA analysis were plotted in Figure 7 and compared.

The discriminating powers of SIMCA demonstrated one particular band with the highest discriminating power in all four cases of analyses, located at 1472 nm, corresponding to water molecules with three hydrogen bonds. The influential bands from all four models are similar, especially in the area of weakly hydrogen bonded water, however, with a different degree of influence in different models. The differences in discriminating powers are mostly in the area of hydrogen-bonded water and strongly bound water, which indicates differences in the water bound to cement components, i.e., the cement matrix differences as well.

It is also interesting to mention the difference in the magnitude of discriminating powers—in the case of W_shallow_ and W_mix_, the magnitude of discriminating powers does not cross 200, while in comparison, discriminating power values for W_dist_ and W_tap_ are about 500 and higher. The influence of the bands is rather uniform for all indicated bands in the case of W_shallow_ and W_mix_, which indicates balanced changes over time in mortar created using these waters.

Next, SIMCA analysis was performed with another objective. The dataset was split according to the day of acquisition of the spectra to Day0, and after 1, 3, and 7 days and data were analyzed by SIMCA to discriminate between mortars prepared using different mixing waters. Despite rather small values of interclass distances between the classes, the discrimination of different mortars according to the age was successful, with a discrimination accuracy higher than 82.64% (Table 4).

In this case also, the discriminating powers of SIMCA analyses (Figure 8) were inspected for the most influential variables. The inspection of discriminating powers demonstrated which absorbance bands were most important for the discrimination between mortars on each of the examined days.

On the Day 0, there are many influential bands, demonstrating lots of differences in the water structure immediately upon mixing: particularly important are the bands at 1490 nm and 1509 nm, which can be attributed to water molecules with four hydrogen bonds and strongly bound water. Moreover, three important bands appear in the area of strongly bound water at 1552, 1571, and 1589 nm. In general, bands above 1500 nm can be assigned to the 1st overtone of the ice-like clusters of water, highly organized molecular structures expected around hydrated macromolecules [83,84,85]. Fujimoto et al. 2007 reported the similar, very closely located bands, 1548 nm and 1592 nm, to be related to crystalline regions in the cellulose matrix of the wood [77]. In the area of weakly hydrogen-bonded water, bands 1323 nm, which can be assigned to water monomers or bulk water [86,87], and band 1379 nm, which corresponds to solvation shells of water, appear significant. On subsequent days, the most differences between the mortars prepared with different water types could be explained by the differences at 1397 nm—trapped water depending on the ion concentration and 1410 nm—free water species that increase in water with the increase in temperature. These two bands become dominant for discrimination, especially after 7 days.

#### 2.3.4. Aquagrams of Cement Mortar

Based on the entirety of the previous analysis, the absorbance bands that appeared consistently and had importance in the interpretation of results were summarized and 18 of them had been selected for representation on aquagram, to describe succinctly the water spectral patterns of cement paste and mortar over the time of investigation and depending on the water used for mixing. These absorbance bands and their tentative assignments with some remarks are provided in Table 5.

The calculated aquagrams presented in Figure 9 and Figure 10, using these 18 bands as radial axes, give a comparative overview of water spectral patterns (WASPs) of cement paste and mortar. First, aquagrams shown in Figure 9 present how WASPs for each cement mortar evolves over time for each mixing water separately, while aquagrams in Figure 10 show how WASPs of mortars created using different waters compare at a particular cement age (days).

From the aquagrams presented in Figure 9, the common trend can be observed for changes from cement paste and mortar with time regardless of water used for mixing. In general, the spectral pattern of water in cement paste is located in the right part of aquagrams, ending at 1472 nm, and with time it moves to the left part, demonstrating that the process of change in the cement during curing is characterized by the transformation of proton hydrates, solvation shells, free and quasi-free water, hydration water, and some small water clusters bonded with 1 to 3 hydrogen bonds. The differences in the cement paste can be better observed from Figure 10a, where it can be observed that W_mix_ water and W_dist_ provide more proton hydrates in cement paste (1342–1354 nm); W_mix_ water seems to provide more free and weakly bound water species (1379–1428 nm), while W_shallow_, in particular, results in more hydrogen-bonded water clusters.

The change in WASPs from cement paste (Day 0) and cement mortar aged 1 day, seem to be particularly different between the waters. During this 24 h, the transformation of the water molecular network is radical, where the major water species reduced during this process are proton hydrates and solvation shells (1342–1379 nm). However, in the case of W_shallow_ and W_mix_ water (Figure 9c,d), these species are lost at the expense of the increased amount of water bonded to the elements of cement matrix (1503–1559 nm), while in the case of W_tap_ and W_dist_, there is a increase in free water, hydration water, solvation shells/surface water, and small water clusters (1410–1460 nm).

The aquagrams comparing the state of mortar after 1 day given at Figure 10b especially demonstrate this difference in the area of strongly bound water. The water species that are initially lost (1342–1379 nm) are the species with the highest energy and highest mobility, such as water vapor, or the so-called moisture [72].

On the other hand, among the bands of strongly bound water, a 1534 nm absorbance band of water was demonstrated to carry the information about the specific volume [113]. The drying shrinkage as a phenomenon is closely related to the moisture loss and to the changes in volume; thus, our findings may indicate the possibility to directly measure those properties and at the same time relate them to the specific water spectral patterns.

Another interesting feature that can be observed from Figure 9, is that the spectral pattern of mortar in the case of W_shallow_ does not change at all in the region of 1385–1447 nm. This lack of change in the cement during first 24 h agrees with what was earlier observed during the PCA analysis. Since the bands 1391 nm, 1397 nm, 1428 nm, and 1447 nm are all related to water species that are in some kind of interaction with the material—either confined, or participating in hydration, solvation, or adsorption, this may indicate that mortar made with W_shallow_ does not really change some elements of the created cement matrix during this time. W_tap_ and W_dist_ mortars, particularly, demonstrate higher absorbance after 1 day at these absorbance bands, corresponding to the absorbance of water species that have less mobility and energy compared to the moisture, but still with lots of ability to participate in chemical reactions.

The WASPs at later days (after 3 and 7) demonstrate quite similar profiles for all mortars, with increasing absorbance at the bands of hydrogen-bonded water and water bonded to elements of cement (1470–1559 nm). This type of water can be considered to be an integral part of the cement and cannot be changed or lost to drying, unless the cement matrix itself is damaged. This agrees with the observations about the non-evaporable water content differences—it was reported that they become less pronounced in three types of Portland cement (different chemical composition of cement) at later ages, and further that the relationship between the non-evaporable water and degree of hydration appears to be dependent upon the chemical composition [118]. Therefore, even though the cement components did not change at all, it may be that the chemical content and hence water molecular structure of the mixing waters contributed to this.

When it comes to the loss of this strongly bound water, as already said, it might happen only due to the damage of the matrix that is binding it, and this may be the case with the mortar made with W_tap_, where, in contrast to all other mortars, the WASPs for the 3rd and 7th day are reversed, with lower absorbance corresponding to the 7th day. This may be the case of cracks in the cement internal structure, where the elements of cement matrix are being broken and the water which was bound, is being released. This can explain the increase in free water that can be observed in Figure 10d, where the highest absorbance in the region 1397–1460 nm can be observed for exactly W_tap_ mortar.

In summary, from the aquagrams, it was observed that WASPs in the first 24 h could be divided in two groups, where W_shallow_ and W_mix_ water mortars were similar, and W_tap_ and W_dist_ mortars were similar. This may result in similar properties of the mortars on the macroscale. W_shallow_ mortar was particularly specific in the terms that it demonstrates stability during first 24 h, while W_tap_ mortar was specific in the terms that it demonstrated opposite change between the 3rd and 7th day of curing. These results, which demonstrate differences at such a detailed scale in the water molecular matrix of the paste and mortar, practically from the very start when paste is mixed, with further research may result in the prediction of mortar properties at the earliest possible stage.

### 2.4. Characterization of the Internal Temperature Change and Thermal Strain in Cement Paste

In ordinary Portland-hydrated cement paste there are four major compounds: tricalcium silicate (C_3_S), dicalcium silicate (C_2_S), tricalcium aluminate (C_3_A) and tetracalcium aluminoferrite (C_4_AF); both C_3_S and C_2_S react with water (H) to form calcium silicate hydrate (C-S-H) and the portlandite, also called calcium hydroxyde (CH) [119,120].

The process of cement hydration is an exothermic reaction and the temperature rise in mass concrete pours [121]. The measured values of internal temperature in cement pastes made by W_tap_, W_dist_, W_shallow_, and W_mix_ water are presented in Figure 11a and demonstrated that this temperature increase is similar for all pastes up to 24 h after casting. The well-known stages of hydration in Portland cement can be observed: (I) initial reaction, (II) period of slow reaction, (III) acceleration period, and (IV) deceleration period. Generally, the hydration reaction starts when cement comes into the contact with water, and the internal temperature of the cement paste rises for a certain period as the cement hardens. However, for the first few hours immediately after casting, the cement paste goes through the setting process where the fluidity is gradually lost, which is different from the above-mentioned hardening process. As shown in Figure 11a, the internal temperature of the cement paste rises and then drops immediately after casting, and then after about 2 h it starts rising again.

There are distinctive differences in the values of internal temperature of paste in the phases I and II, depending on the water used for preparation. In phase I, the paste made with W_dist_ shows the fastest increase and the highest temperature during the initial reaction of cement with water. In phase II, the temperature values are quite different between all the pastes, but in particular between the cement made with W_dist_ and W_mix_ water, demonstrating differences in this stage of cement hydration with respect to the water used. Important point to take into consideration is that the internal temperature at the start of measurement is already different for each cement paste. Therefore, to be able to grasp real change in the internal temperature compared to the initial temperature of the paste, the temperature difference during phases III and IV is plotted in Figure 11b.

The results shown in Figure 11b indicate that the amount of released heat is at its maximum 12.25 h after casting for W_mix_ water, 12.8 h for W_dist_ and W_shallow_, and 13.25 h for W_tap_. This suggests that the progression of hydration reaction happens at a different speed depending on the water used, and it is especially different in the case of W_tap_. The maximum heat generation was the largest for the W_dist_, followed by W_shallow_, W_tap_, and W_mix_.

The hardened cement paste has different thermal expansion properties, which leads to different volumetric changes that generate internal stress causing the cracking of concrete at the micro- or macro-scale [90,122,123]. The thermal expansion coefficients (TEC) are affected by the water contents in a hardened cement paste, especially the free water content, which is why TEC have large values in the early hydration ages [124]. The TEC of a compacted material is smaller than that of a porous material [125].

The measured values of thermal expansion during 24 h after casting are plotted in Figure 12. The thermal strain was larger in the cases of W_dist_ and W_tap_, which were quite similar, but different compared to the thermal strain recorded in mortar made by W_shallow_ and W_mix_ water (Figure 12). The grouping strongly resembles grouping two-by-two that can be observed in the WASPs of mortars after 1 day (Figure 10b). Given the above background, it can be expected that hardened paste created by W_shallow_ and W_mix_ water have a less porous, more compact structure.

### 2.5. Characteristics of Drying Shrinkage Strain

Drying shrinkage is defined as the volumetric change of concrete induced by the loss and redistribution of moisture, which can lead to the formation of cracks within the concrete and influence important properties such as durability, deformation, and stress distribution [126,127,128]. It is therefore desirable to reduce drying shrinkage; the techniques proposed so far are to apply low shrinkage cement or cement with low hydration heat [129]. It is also reported that drying shrinkage is more pronounced in specimens with rapid moisture loss [130]; therefore, slowing down this process could help in the reduction of drying shrinkage. As this study already pointed out, this may be achieved by the adequate choice of water for the preparation of cement.

The results of the drying shrinkage strain measurements performed over the period of 91 days after casting demonstrate considerably reduced drying shrinkage strain in specimens prepared with W_shallow_ and W_mix_ water (Figure 13a). The worst performance can be observed in cement prepared with the W_tap_, followed by W_dist_. The drying shrinkage strain properties already demonstrated the same result even only 7 days after casting (Figure 13b). The drying shrinkage strain was about the same for W_shallow_ and W_mix_ water, and W_dist_ and W_tap_ followed in increasing order. Similar to the results of thermal stress, grouping two-by-two can also be observed here in Figure 13, which closely matches the pattern observed in WASPs of cement paste given in Figure 10b.

To understand the reason why the drying shrinkage strains of W_shallow_ and W_mix_ water were smaller, the results of the water component analysis and spectral patterns of cement paste and mortar were taken into consideration.

First, the water composition analysis demonstrated that the Si content was considerably higher in W_shallow_ and W_mix_, compared to the W_dist_ and W_tap_, as shown in Table 1 and Figure 1. W_shallow_ and W_mix_ water also contained a large amount of sodium. Generally, when Ca(OH)_2_ produced by the hydration reaction of cement comes into contact with silicate compounds such as Na_2_O·SiO_2_ at the initial stage, CaO·SiO_2_ crystals are formed. This is understood to improve the watertightness of the hardened cement, and the application of silicate compounds is used for improving the properties of cement concrete [131,132,133]. In this study, it can be concluded that Si and Na in the W_shallow_ and W_mix_ water affected the crystal structure of cement mortar, and the suppression performance against drying shrinkage might have improved as a result. The water spectral patterns of cement paste 24 h after setting, indeed demonstrate an increase in strongly bound, crystalline water (Figure 10b, region 1490–1559 nm). The spectral patterns of mortar 7 days after casting also demonstrate the same for W_shallow_ and W_mix_ mortars compared to the W_dist_ and W_tap_ mortar (Figure 10d).

Further, since W_shallow_ demonstrated the best drying shrinkage characteristics, it is of interest to notice that the spectral pattern of cement paste after 24 h (Figure 10b) was particularly different, demonstrating no changes in absorbance in the region of 1385–1447 nm, while the W_mix_ paste demonstrated no changes in absorbance in the narrower region of 1441–1447 nm. In contrast, the cement paste created with W_tap_ and W_dist_ demonstrated an increased amount of water species absorbing in these regions. The water species that are initially lost (1342–1379 nm), the so-called moisture [72], in the case of W_shallow_ and W_mix_ water (Figure 9c,d), led to the increased amount of water bonded to the elements of cement matrix (1503–1559 nm), while in the case of W_tap_ and W_dist_, to increase in free water, hydration water, solvation shells/surface water, and small water clusters (1410–1460 nm). Considering the resulting drying shrinkage properties, this indicates that this particular water molecular structure might be especially important to monitor during the early age of cement. In this respect, this study has made considerable progress in pinpointing the exact absorbance bands corresponding to specific water structures within cement paste and mortar that can serve as WAMACs, i.e., the coordinates at which the absorbance of water in concrete can be measured and provide the information about the state of water within cement matrix and how it is related to the current state and future properties of produced concrete.

To conclude, it is clear that even a small difference in the water composition used for the cement mortar affects the molecular structures of the water inside the cement mortar and lead to different mechanical properties. The difficulties of relating single components to a specific function were overcome by using water spectral pattern as an integrative multidimensional marker, displayed on aquagram and mirroring even a small perturbation to the water molecular matrix in each mixture. These research findings may be interpreted within the existing framework of current understanding, but for the first time the state of the water in cement was described using a water spectral pattern, where numbers were used instead of the vague and broad terms such as capillary water, adsorbed water, interlayer water, and others. This provides a strong basis for the development of a universal method for the characterization of cement paste, mortar, and concrete from an aspect of hydration and allows quantitative comparisons and the optimal choice of cement materials and water for mixing with possibilities for early prediction of cement quality by using non-invasive monitoring.

## 3. Materials and Methods

### 3.1. Water Samples

Four types of water were used for the preparation of the cement: tap water from the Osaka city municipal supply, distilled water, and two spring mineral waters (Table 6). Mineral spring waters were first filtered using activated charcoal to remove the impurities, and then using an antibacterial filter.

### 3.2. Cement and Fine Aggregate (Sand)

For the preparation of cement, Ordinary Portland cement (OPR) was used. The main components are CaO, SiO_2_, Al_2_O_3_, and Fe_2_O_3_, but since the content ratio differs depending on the manufacturing plant, the quality of Portland cement is standardized by JIS (Japanese Industrial Standards) R5210 “Portland cement”. As an example, Table 7 shows the standard for the chemical composition contained in cement. The density of the cement used was 3.15 g/cm^3^. Australian standard sand conforming to ISO 679 was used as the fine aggregate. Table 8 shows the chemical composition of standard sand, and Table 9 shows an example of its physical properties, respectively.

### 3.3. Preparation of Hardened Cement Specimens

The specimens were prepared at the Osaka Metropolitan University (formerly Osaka City University) Graduate School of Engineering, Department of Urban Design and Engineering. The specimens produced were cement paste and cement mortar. The mixing ratio of cement paste and cement mortar was determined according to JIS R 5201, the “Physical testing methods for cement” [134].

Cement paste is a mixture composed of water and cement with a mass ratio of water to cement of 1:2. Cement mortar is a mixture composed of water, cement, and fine aggregate, and the mass ratio of water, cement, and sand is 1:2:6. In order to minimize the influence of temperature when mixing the materials, water, cement, and fine aggregate were left in a curing room at the constant temperature one day before the mixing of materials. The mixing of cement paste and cement mortar was carried out in accordance with JIS R 5201, the “Physical testing methods for cement” [134], as follows:Add water and cement to the container for mixing.Mix at a low speed for 30 s.Add fine aggregate.Mix at a high speed for 30 s.Scrape the mortar adhering to the walls and bottom of the mixing container (Stop for 90 s).Mix at high speed for 60 s and then take out.

For cement paste, the procedure ends with step 2. The cement paste was left in a plastic beaker for subsequent measurements (Figure 14). After casting in steel formwork, the surface of the cement mortar specimens was covered with a wrap, and the specimens were left in a curing room at a constant temperature of 20 °C ± 2 °C and humidity of 60 ± 5%. The formwork was removed the next day, and the specimens were placed in the curing room at the constant temperature and in the air atmosphere.

### 3.4. NIR Spectroscopy

#### 3.4.1. NIR Spectroscopy Measurement of Water Samples

The water used in this study was analyzed at the Aquaphotomics Research Department, Graduate School of Agricultural Science, Kobe University. The NIR spectra of the water samples were measured with the FOSS XDS spectrometer equipped with a Rapid Liquid Analyzed (RLA) module (FOSS NIRSystems, Inc., Höganäs, Sweden) using a quartz cuvette (optical path length 1 mm). The spectra were acquired in the transmittance mode, with the resolution of 0.5 nm in the spectral range 400–2500 nm. Each recorded spectrum was an average of 32 co-added spectra. The water samples were prepared in triplicates, and each sample was measured with 5 consecutive scans. The experiment was performed at 25 °C, keeping the temperature of the sample holder at this temperature throughout the measurements using circulating water bath. The order of measurements was randomized, but after every 5 samples, the sample of ultra-pure water (MilliQ, Millipore, Molsheim, France) was scanned. The temperature of the samples was monitored and logged during the experiment.

#### 3.4.2. NIR Spectroscopy Measurement of Cement Paste and Mortar

Measurement of cement specimens was carried out at the Graduate School of Engineering, Department of Urban Design and Engineering, Osaka Metropolitan University. Using a portable, handheld NIR spectrometer (MicroNIR OnSite-W, VIAVI Solutions Inc. Milpitas, CA, USA), the spectrum was measured in 6.2 nm increments in the wavelength range of 950 nm to 1650 nm. The measurement was taken on Day 0 (immediately after mixing, cement paste), 1 day, 3 days, 7 days, and 28 days after mixing (cement mortar).

The NIR spectrometer was fixed as shown in Figure 14 to measure the specimen immediately after mixing. The measurements were performed with a gap of approximately 10 mm between the cement paste surface and the NIR spectrometer. The mortar specimens were measured as presented schematically in Figure 15 at particularly chosen measurement points. The spectral measurements were acquired at 5 points on the right and left surface of the specimen, with 3 consecutive measurements each. These measurements were performed 1 day, 3 days, 7 days, and 28 days after mixing.

### 3.5. Aquaphotomic Spectral Data Analysis

#### 3.5.1. Water Characterization

The characterization of mixing waters was performed using aquaphotomics NIR spectroscopy. The NIR spectra of mixing waters, acquired in the range 400–2500 nm, at controlled temperature of 25 °C were trimmed to region of 1300–1600 nm that corresponds to first overtone of water stretching vibrations which typically has a maximum around 1450 nm.

The spectra were analyzed according to the protocol of aquaphotomic spectral analysis [63]. Difference spectra, Principal Component Analysis (PCA) [64], Soft Modeling of Class Analogies (SIMCA) [65] and Partial Least Squares Regression (PLSR) Analysis (using temperature and consecutive irradiation as dependent variables) were performed (results not presented) in order to find the representative water absorbance bands—water matrix coordinates (WAMACS) [51] that can be used to depict the characteristic water spectral patterns (WASPs) of mixing waters on aquagrams [63,66]. The selection of WAMACS was performed as described in the recent literature [63,67], by choosing the consistently repeating and most influential absorbance bands in the entire performed analysis. This resulted in selection of 15 wavelengths to serve as WAMACS and create aquagrams.

The aquagram shows the average normalized absorbance of the spectra of various samples at selected 15 wavelengths associated with the various molecular structures of water which define radial axes of the graph. Microsoft Office Excel 2013 (Microsoft Co., Redmond, WA, USA) was used for aquagram calculations.

#### 3.5.2. Cement Paste and Mortar Characterization

The spectral data were converted to pseudo-absorbance (logT^−1^ where T is the transmittance). In order to investigate the first overtone of the OH stretching vibration, the wavelength range of the measured spectrum was limited to 1300 to 1600 nm.

The exploratory analysis in the form of Principal Component Analysis (PCA) [75] was applied on the spectral data separated in four datasets according to the type of water used for mixing, and corrected for baseline effects using detrend and standard normal variate transformation [135] after the smoothing to eliminate the noise from the spectra (Savitzky-Golay 2nd order polynomial filter [76] and 21-point window size).

Soft Modeling of Class Analogies (SIMCA) [65] is a supervised pattern recognition technique, used in the study to discriminate between the mortars created by different mixing waters and the age of the mortars.

Aquagrams of the cement paste and mortar were prepared following the same procedure as described in the Section 3.5.1, with the difference that spectral data were preprocessed using smoothing, linear detrend and standard normal variate transformation, and 18 wavelengths were selected to be presented on aquagrams.

### 3.6. Physical Test Method for Cement Mortar

All the tests were conducted at the Graduate School of Engineering, Department of Urban Design and Engineering, Osaka Metropolitan University.

#### 3.6.1. Temperature Change and Thermal Strain

The specimens were prepared using cement paste for this test. A strain gauge with a temperature measurement function was used to reduce the influence of surrounding environmental conditions such as temperature. In order to place the strain gauge near the center of the specimen, a jig with the strain gauge attached was installed in the formwork in advance, and then cement paste was placed in the formwork. A schematic representation of the jig with the attached strain gauge is presented in Figure 16a. This jig was fixed to the bottom of a cylindrical formwork with a diameter of 50 mm and a height of 100 mm using an adhesive (Figure 16b). The final formwork for experimental measurements before the cement paste was poured in is shown in photographs in Figure 16c.

The cement paste was then placed in formwork and left in the air in a constant temperature curing room with a temperature of 20 ± 2 °C and a humidity of 60 ± 5%.

The measurement of the thermal change of the cement paste was started immediately using the digital micron strain gauge, contact type (Mitutoyo ABSOLUTE, Mitutoyo Corporation, Kawasaki, Japan). The measurements were performed every 15 min until approximately 24 h after the cement paste was placed.

#### 3.6.2. Dry Shrinkage Test

The drying shrinkage strain test was performed in accordance with JIS A 1129-2 “Methods of measurement for length change of mortar and concrete—Part 2: Method with contact-type strain gauge.” The measuring instrument used was the digital micron strain gauge (contact-type gauge), shown in Figure 17.

The specimens were produced using a regular right prism formwork with dimensions of 40 mm × 40 mm × 160 mm. After taking them out of the formwork, a gauge plug was attached to the side of each specimen. Figure 17 shows the position of the gauge plug attachment. The specimens were then left to stand in the air in a constant temperature curing room with a temperature of 20 ± 2 °C and a humidity of 60 ± 5%.

This test was performed with the aim to calculate the rate of the shrinkage of the specimen using the measured length between the gauge plugs. The measurements of the length between the gauge plugs were performed on the day 1 (when taking the specimens out of the formwork) and 3 days, 5 days, 7 days, 14 days, 21 days, 28 days, 42 days, 56 days, and 91 days after casting. The rate of the length change was calculated using the following equation:(1)$$\varepsilon =\frac{X_{i}-X_{0}}{L_{0}}\times 10^{6}$$
where ε is the drying shrinkage strain or change in length (×10^−6^), L_0_ is the base length (length between gauge plugs), X_i_ is the measured value of the specimen i day(s) after casting, and X_0_ is the measured value of the specimen immediately after casting.

## 4. Conclusions

This research study conducted an aquaphotomic near infrared spectral analysis to investigate the effects of the different types of water on the shrinkage characteristics of the hardened cement. The aquaphotomic characterization of mixing waters and monitoring of cement mortar was performed immediately after mixing, and after 1, 3 and 7 days of curing in air. The change in internal temperature and thermal strain in cement paste were measured during first 24 h after mixing, and the dry shrinkage strain was measured up to the period of 91 days.

The research results demonstrated that the measured mechanical properties of the hardened cement material differed depending on the water used. Specifically, based on the results of the analyses, the following conclusions were drawn:

The results of the standard analysis of mineral constituents in four mixing waters demonstrated small differences considered negligible and irrelevant for the cement production according to the standards for the most countries. However, aquaphotomics’ characterization demonstrated that mixing waters have a higher solvation ability compared to the pure water and the largest differences between the four types of waters were found at the 1448 nm water absorbance band, assigned to the absorption of water molecules in solvation shells with four or five water molecules.The PCA analysis of cement paste and mortar created by different mixing waters demonstrated that the major variation in the spectra can be described by only two principal components, related to the changes of cement mortar during curing (in terms of days) and to the changes during early hydration reaction in the first 24 h. The most important water absorbance bands for the description of changes during curing were identified at 1391, 1397, 1472, and 1478 nm. The first two can be attributed to the absorbance of the confined water molecules in the interlayer between the crystal lattice, while the latter two to water molecular species with three hydrogen bonds indicated the semi-crystalline state of cement. For the description of the initial hydration reaction, the most important absorbance bands were found at 1360 and 1366 nm assigned to water solvation shells around ions, located at the edge and the corners of crystal lattices, and at 1472 nm, the water molecules bonded with three hydrogen bonds. There are indications that first two bands could be related to the hardness, which agrees well with the understanding of cement curing as the process of hardening of cement. The process of cement curing was described as a transformation of weakly hydrogen-bonded water to hydrogen-bonded water, which agrees with the release of heat during the reaction of cement hydration. Despite the common absorbance bands present in developed PCA models, each cement mortar demonstrated a specific time evolution depending on the water used for its preparation.The results of the SIMCA discriminating analysis confirmed that it is possible to discriminate the age of cement mortar with an accuracy higher than 90%, and to discriminate between mortars made with different mixing waters with accuracy higher than 82%. The discriminating powers of SIMCA demonstrated the importance of the absorbance band of 1472 nm (water molecules with three hydrogen bonds) for discrimination. The differences were found mostly in the area of hydrogen-bonded water and strongly bound water, which indicates differences in the water bound to cement components, i.e., the cement matrix differences as well.The entirety of aquaphotomics analysis discovered 18 water absorbance bands: 1342, 1354, 1366, 1379, 1385, 1391, 1397, 1410, 1428, 1441, 1447, 1460, 1472, 1490, 1503, 1515, 1534, and 1559 nm as absorbance bands that could be used to measure the state of water directly and the state of cement mortar during curing indirectly, over time. These absorbance bands can be considered as WAMACS, i.e., Water Matrix Coordinates and their combination was used to depict Water Spectral Patterns—WASPs of cement mortar in aquagrams. The aquagrams revealed that W_shallow_ and W_mix_ water mortars were similar, and W_tap_ and W_dist_ mortars were similar, indicating similar properties of the mortars on the macroscale. The aquagrams demonstrated differences at such a detailed scale in the water molecular matrix of the paste and mortar, practically from the very start when paste is mixed, providing a possibility for the prediction of mortar properties at the earliest possible stage.The measured values of thermal strain revealed that W_shallow_ and W_mix_ water mortars were similar, and W_tap_ and W_dist_ mortars were similar, strongly resembling grouping two-by-two, which is observed in the WASPs of mortars. Judging by the WASPs, it was concluded that hardened paste created by W_shallow_ and W_mix_ water has a less porous and more compact structure. The results of the drying shrinkage strain measurements performed over the period of 91 days after casting demonstrate considerably reduced drying shrinkage strain in specimens prepared with W_shallow_ and W_mix_ water. The drying shrinkage strain was about the same for the W_shallow_ and W_mix_ cement mortar, and W_dist_ and W_tap_ followed in increasing order. Similar to the results of thermal stress, grouping two-by-two was also observed in drying shrinkage properties closely matching the pattern observed in WASPs of cement paste.

In summary, using aquaphotomics, it was demonstrated that four waters used for cement mixing had significantly different molecular structure, which influenced the cement hydration and curing, causing differences in water molecular dynamics after casting that resulted in different mechanical properties, specifically, in thermal and drying shrinkage of produced cement mortar. The mineral waters could be considered as a better choice to be used in the mixing of cement because they provide better shrinkage behavior.

Understanding the mechanisms of cement hydration intersects both scientific and practical interests. This study is, in this sense, a pioneering one, since for the first time, a novel, completely non-invasive, non-destructive, and rapid method based on aquaphotomics is presented and evaluated for this purpose. From scientific aspects, this research presented a novel way to describe the chemical and microstructural phenomena that characterize cement hydration by directly following the changes in the molecular structure of water within the cement, in a very detailed manner, with defined water molecular species and their functionality explained. From practical aspects, this study demonstrated the need to update the current standards regarding the water that can be used for mixing concrete, by demonstrating the impact the molecular structure of water has on shrinkage behavior. Further, this also provides a basis for development of a precise quantitative method that allows for rapid assessment and comparisons of the cement concrete at the very place where the production is performed.

Finally, this study is not without limitations: the focus was only on investigating four types of water and how they are related to shrinkage properties. The shrinkage is not the only concerned property to evaluate the overall performance of cement concrete, and further aquaphotomics studies will be directed at also evaluating strength and permeability. Another limitation is that the link between the water spectral patterns and the resulting mechanical properties was made primarily qualitatively, by comparison. This can be overcome by better experimental design, with spectral acquisition and reference measurements performed simultaneously, which will allow the development of quantitative prediction models and discovery of direct correlation patterns between particular water species and the measured mechanical properties.

## Figures and Tables

**Figure 1 molecules-27-07885-f001:**
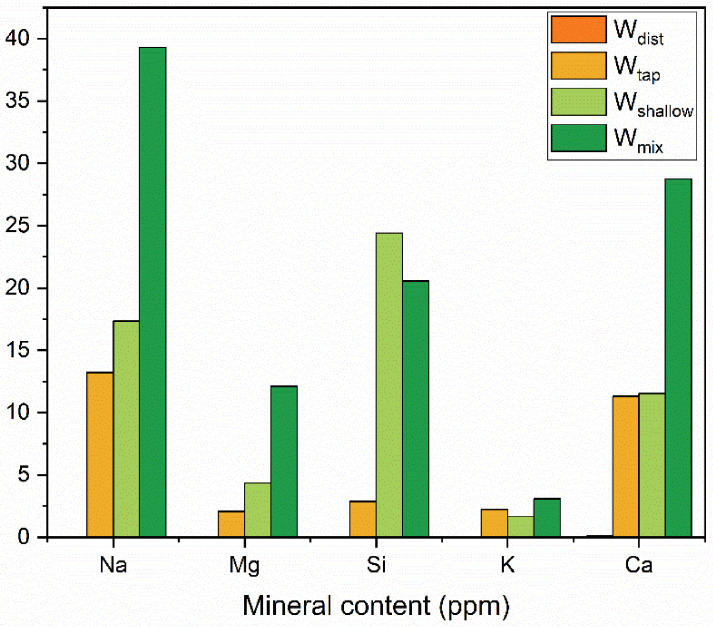
Major differences in mineral content of waters used for cement preparation.

**Figure 2 molecules-27-07885-f002:**
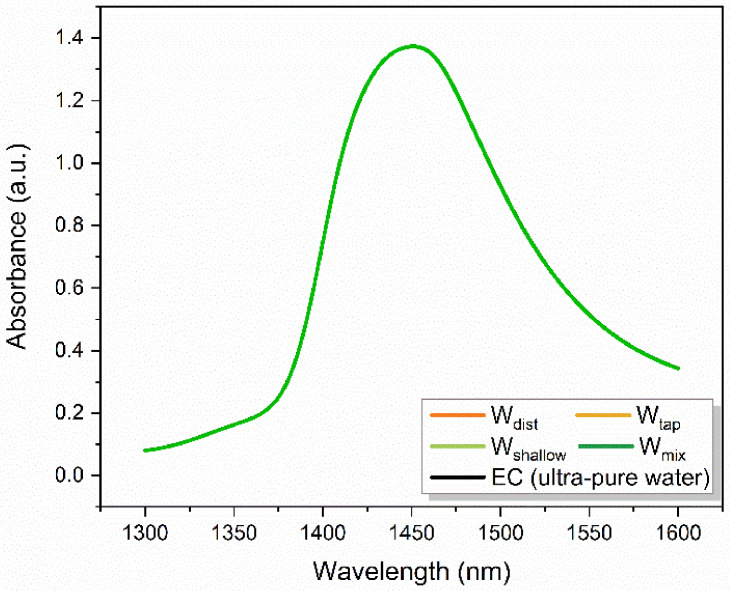
Raw absorbance spectra of mixing waters used for preparation of cement.

**Figure 3 molecules-27-07885-f003:**
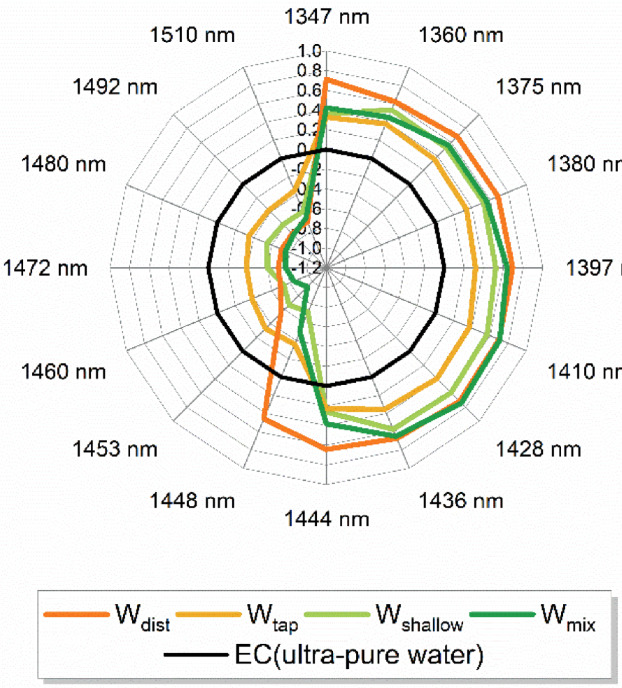
Aquagrams of various mixing waters used for preparation of cement.

**Figure 4 molecules-27-07885-f004:**
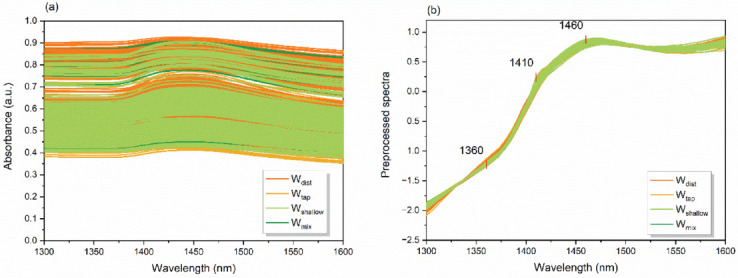
Near infrared spectra of cement paste and mortar samples created using different waters: (**a**) Raw absorbance spectra; (**b**) Preprocessed spectra using linear detrend correction and standard normal variate transformation.

**Figure 5 molecules-27-07885-f005:**
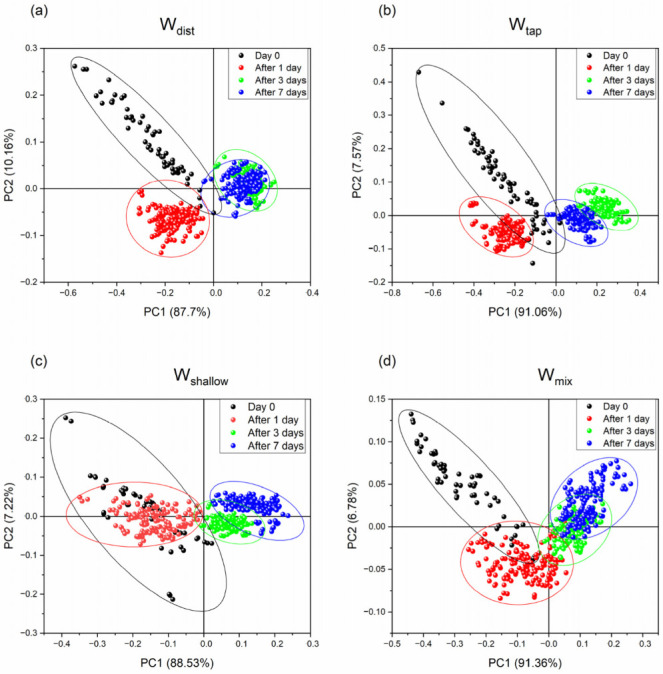
Transformation and the projection of the original spectral data into the space defined by two orthogonal principal components (PC1 and PC2) that retained more than 95% of original variance. PC1-PC2 scores plots of PCA analysis show changes in cement mortar prepared with different mixing water: (**a**) W_dist_, (**b**) W_tap_, (**c**) W_shallow_, and (**d**) W_mix_. According to the results of the analysis, the PC1 axis can be related to the changes in water of cement as the time progresses from Day 0 to 7, while the PC2 axis describes changes in water during first 24 h.

**Figure 6 molecules-27-07885-f006:**
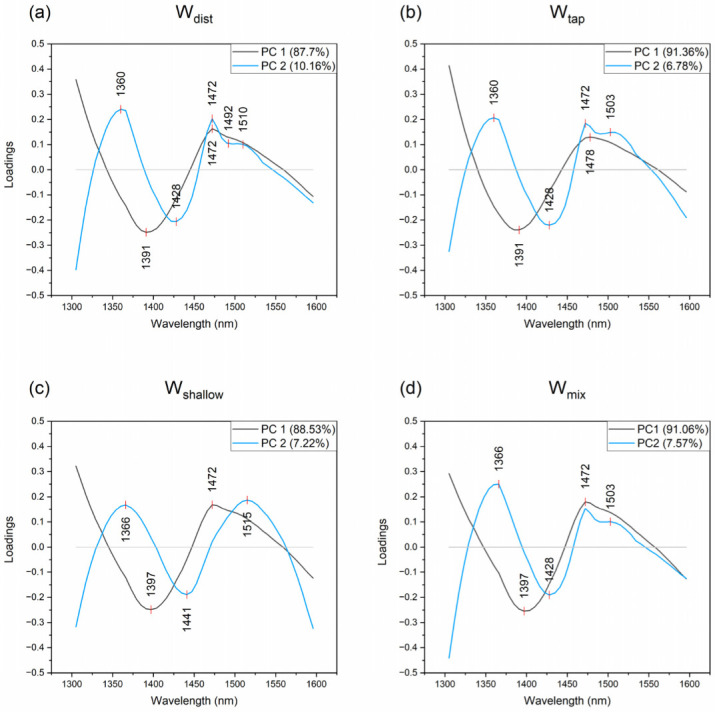
PC1 and PC2 loadings plots of PCA analysis describe changes in the water molecular structure during hardening of cement mortar prepared with different mixing water: (**a**) W_tap_, (**b**) W_mix_, (**c**) W_dist_, and (**d**) W_shallow_.

**Figure 7 molecules-27-07885-f007:**
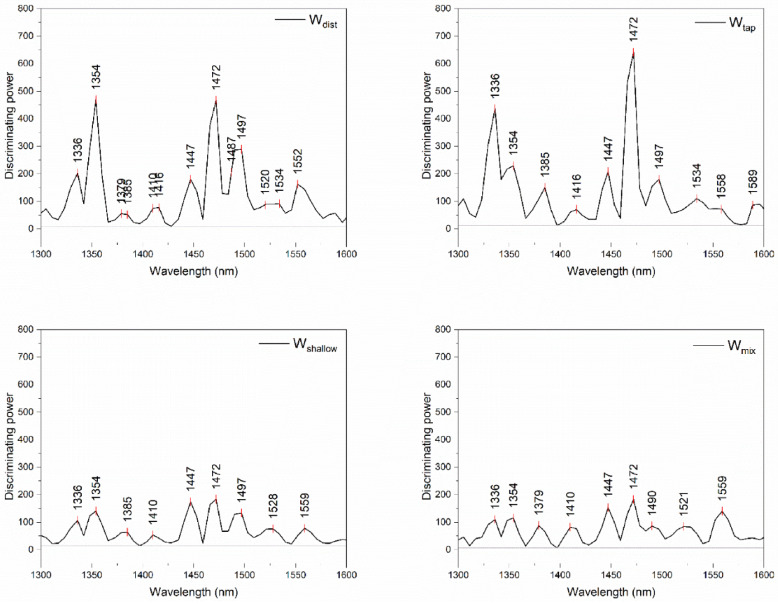
Discriminating powers of SIMCA analysis for discrimination of days when the spectra were acquired from paste and mortar prepared using 4 different waters.

**Figure 8 molecules-27-07885-f008:**
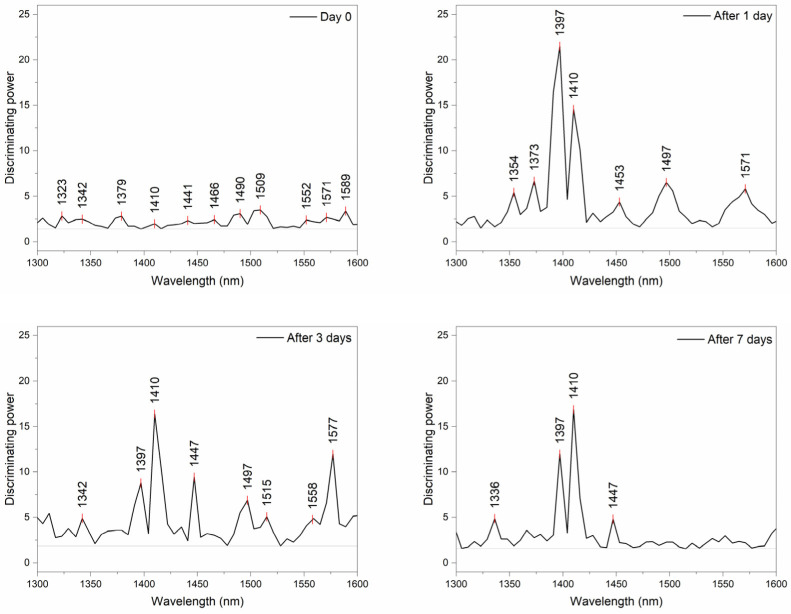
Discriminating powers of SIMCA analysis for discrimination of different mortars, on separate days during the process of setting cement concrete.

**Figure 9 molecules-27-07885-f009:**
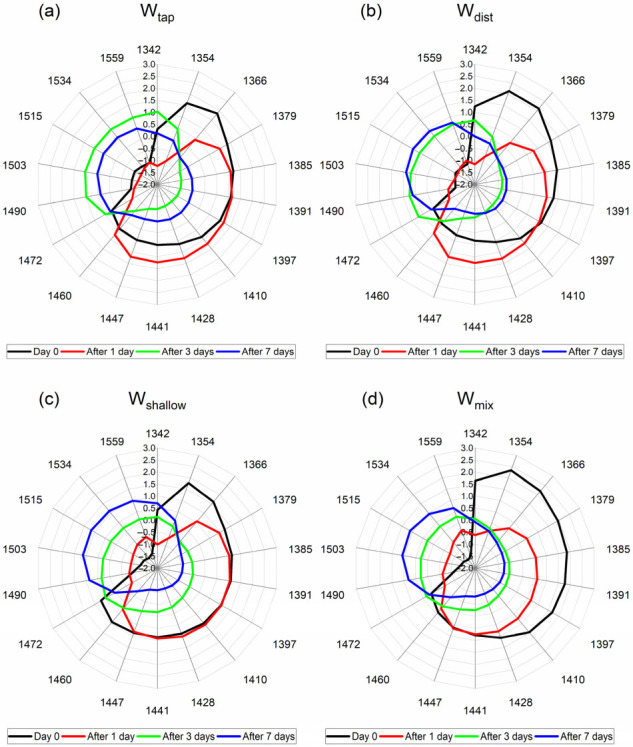
Aquagrams of cement paste and mortar according to the water used for cement mixing: (**a**) W_tap_, (**b**) W_dist_, (**c**) W_shallow_, and (**d**) W_mix_ water. The aquagrams demonstrate the differences in water spectral patterns (WASPs) of cement mortar at different points in time: at Day 0—immediately after mixing of cement paste, and when the mortar was removed from the frame and aged 1 day, 3 days, and 7 days. The aquagrams are colored according to the corresponding measurement day.

**Figure 10 molecules-27-07885-f010:**
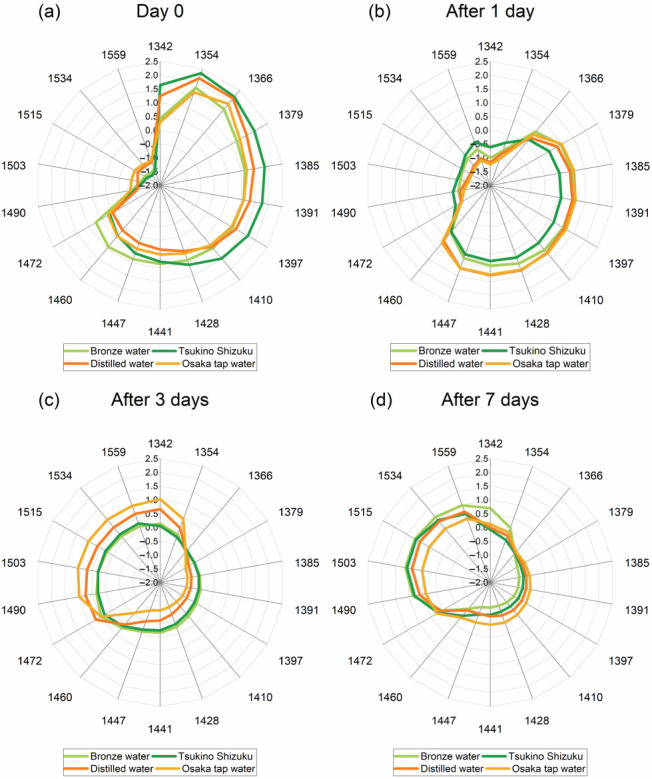
Aquagrams of cement paste and mortar grouped according to the time points when the measurements were performed: (**a**) Day 0—immediately after mixing of cement paste, (**b**) one day after the mortar was removed from the frame and left to cure, (**c**) three days after the mortar was removed from the frame and left to cure, and (**d**) seven days after the mortar was removed from the frame and left to age. The aquagrams demonstrate the differences in water spectral patterns (WASPs) of cement mortar depending on the water used for mixing, and the aquagram profiles are colored accordingly.

**Figure 11 molecules-27-07885-f011:**
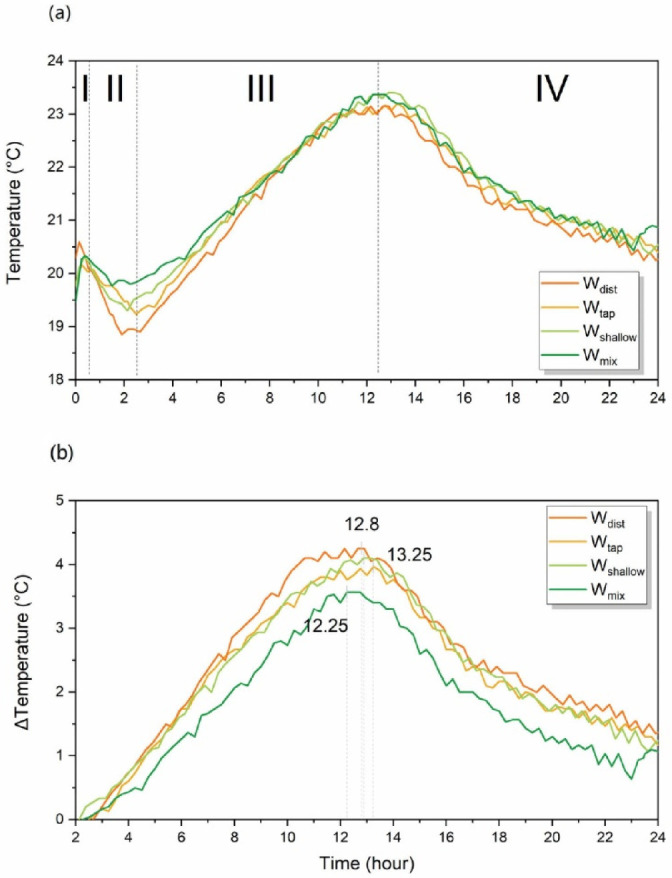
(**a**) Internal temperature of the cement paste specimens during first 24 h; (**b**) the trend of the change in the internal temperature of cement paste during phases III and IV.

**Figure 12 molecules-27-07885-f012:**
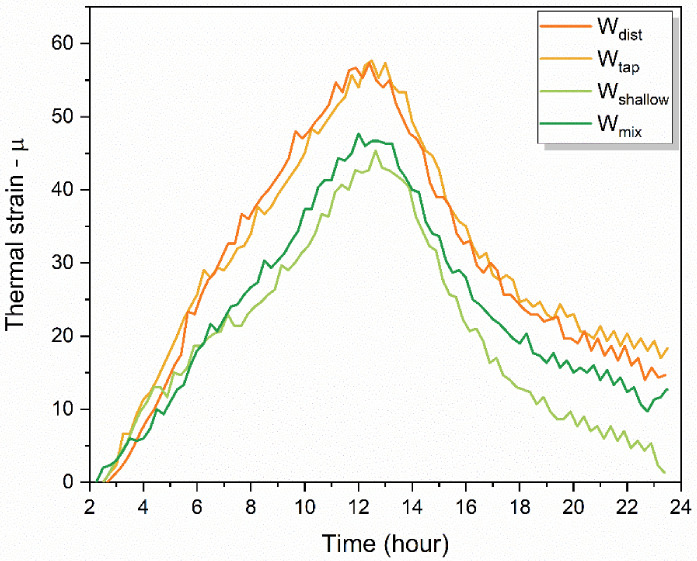
The time trend of thermal strain of cement paste.

**Figure 13 molecules-27-07885-f013:**
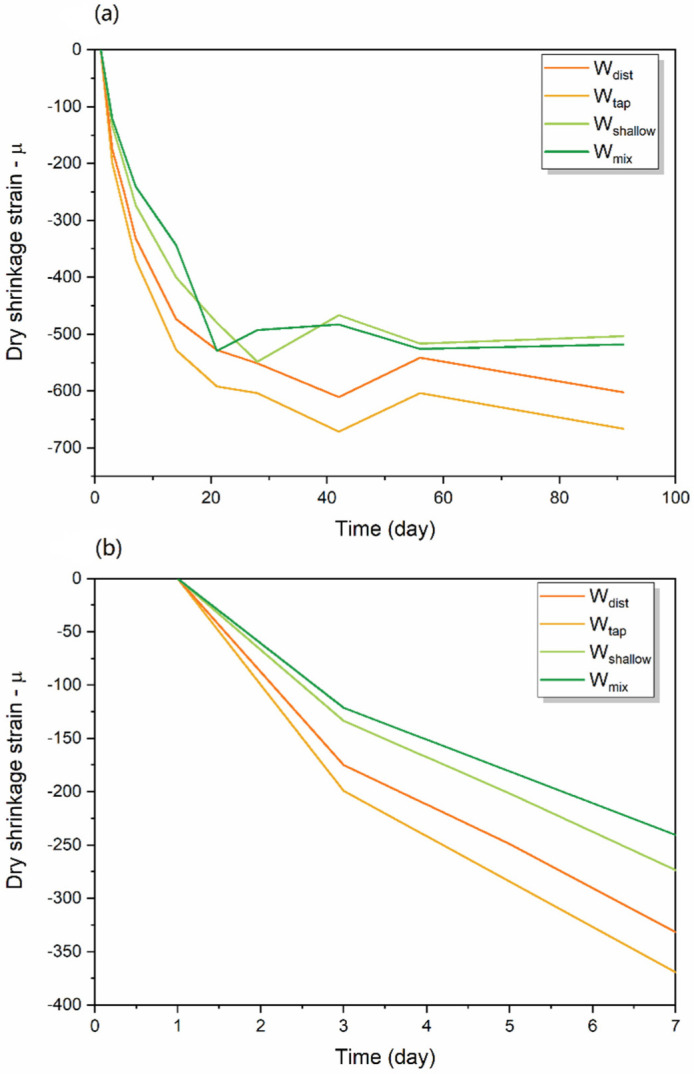
Drying shrinkage strain: (**a**) in time period up to Day 91; (**b**) in time period up to Day 7.

**Figure 14 molecules-27-07885-f014:**
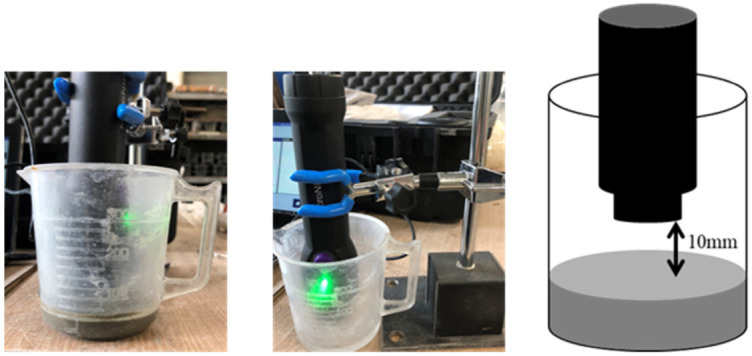
Spectral measurement immediately after mixing (cement paste).

**Figure 15 molecules-27-07885-f015:**
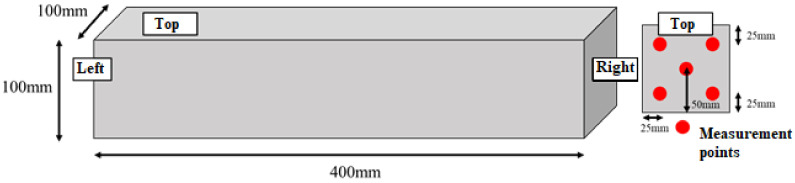
Measurements of cement mortar specimens.

**Figure 16 molecules-27-07885-f016:**
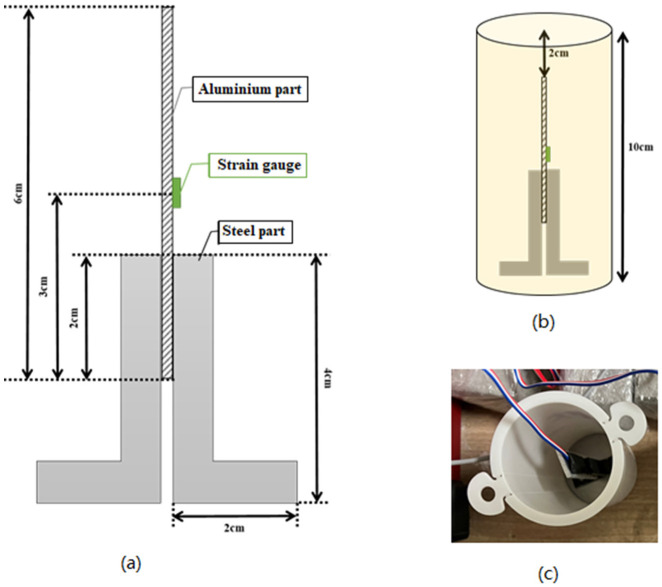
Measurements of the thermal changes of the cement paste; the method of fixing the strain gauge: (**a**) schematic representation of attached strain gauge, (**b**) arrangement of the jig in the formwork, and (**c**) experimental setup.

**Figure 17 molecules-27-07885-f017:**
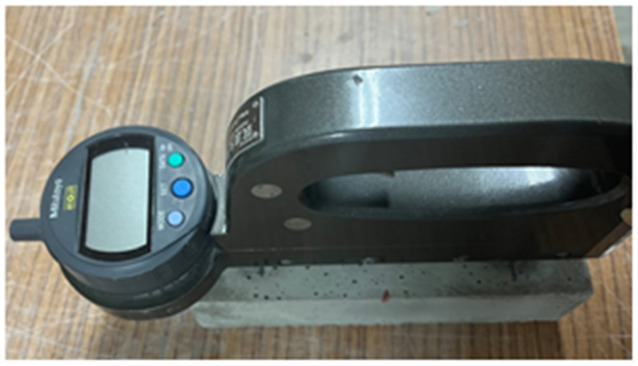
Measurement of length change of mortar using digital micron strain gauge (contact-type gauge).

**Table 1 molecules-27-07885-t001:** Mineral content of the waters used for preparation of cement mortar. W_mix_ water has the highest mineral content, followed by W_shallow_ and W_tap_.

Contained Elements (ppm)	W_dist_	W_tap_	W_shallow_	W_mix_
Li	0.000	0.000	0.000	0.000
B	0.000	0.014	0.057	0.540
Na	0.000	13.220	17.340	39.300
Mg	0.002	2.074	4.364	12.105
Al	0.000	0.014	0.000	0.006
Si	0.000	2.870	24.400	20.570
K	0.000	2.234	1.679	3.094
Ca	0.107	11.315	11.555	28.750
Ti	0.000	0.000	0.000	0.000
Mn	0.000	0.000	0.001	0.108
Fe	0.000	0.013	0.001	0.006
Cu	0.000	0.000	0.006	0.000
Zn	0.000	0.003	0.012	0.002
Sr	0.000	0.526	0.133	0.273
Ag	0.000	0.000	0.000	0.000
Ba	0.000	0.012	0.019	0.046

**Table 2 molecules-27-07885-t002:** The percentage of explained variance by two first principal components from the results of PCA analyses performed on four separate datasets of cement mortar depending on the mixing water.

Mixing Water	PC1 (%)	PC2 (%)	PC1 and PC2 (%)
W_dist_	87.7	10.16	97.86
W_tap_	91.06	7.57	98.63
W_shallow_	88.53	7.22	95.75
W_mix_	91.36	6.78	98.14

**Table 3 molecules-27-07885-t003:** SIMCA analysis results: discrimination accuracy and values of interclass distance compared to Day 0.

Water	Interclass Distance Compared to the Day 0			Discrimination Accuracy (%)
	After 1 Day	After 3 Days	After 7 Days	
W_tap_	10.392	6.342	5.191	91.03
W_dist_	9.511	4.977	6.312	93.53
W_shallow_	9.807	4.539	4.030	94.12
W_mix_	8.945	6.471	7.204	98.18

**Table 4 molecules-27-07885-t004:** SIMCA analysis results: discrimination accuracy and values of interclass distance between mortars prepared by different waters on the day of preparation and after 1, 3, and 7 days.

Day	Interclass Distance Range Min to Max	Discrimination Accuracy (%)
Day 0	0.319–0.606	82.64
After 1 day	0.244–2.056	88.33
After 3 days	0.256–1.589	94.36
After 7 days	0.244–1.013	91.68

**Table 5 molecules-27-07885-t005:** Tentative assignments of the absorbance bands found to be important during analysis of cement spectral data. The wavelengths given in the parentheses in the assignments’ column are the band positions from the cited literature and recalculated from wavenumbers or calculated overtones from fundamental frequencies reported in the original source.

Absorbance Band [nm]	Assignment/Remark
1342	(1342.6 nm, 1st overt. of 3724 cm^−1^) proton hydrates [H+·(H_2_O)_3_]—H_2_O asymmetric stretch, 1st overt. [83] WAMACS C1: 1336–1348 nm: 1st overtone ν_3_ asymmetric stretch [51]
1354	(1353.18 nm, 1st over. of 3695 cm^−1^) two to four nonbonded OH stretches in 2 to 11 member cluster of hydrated proton [83] (1353.55 nm, 1st overt. of 3694 cm^−1^) free OH stretch (OH-·(H_2_O)_2_) [88]
1366	(1366.12 nm, 1st overt. of 3660 cm^−1^)—proton hydrates [H+·(H_2_O)_2_]—H_2_O asymmetric stretch [83] (1366.12 nm 1st overt. of 3660 cm^−1^) OH-stretch in (OH-·(H_2_O)_2_) [88] (1366.1 nm) Dangling -OH (non-hydrogen-bonded), 1st overt. [89] (1362 nm (7339 cm^−1^)) OH-coordinated with 1 or 2 Mg^2+^ on the corner and edge of the Mg(OH)_2_ surface [78] (1368 nm (7306 cm^−1^)) OH-coordinated with 3 Mg^2+^ [78] (1366 nm)—absorbance band of a compound highly correlated with hardness [79] WAMACS C2: 1360–1366 nm—water solvation shell OH-(H_2_O)_1,2,4_ [51]
1379	WAMACS C3: 1370–1376 nm—combination symmetric asymmetric stretch ν_1_+ ν_3_ [51] or WAMACS C4: 1380–1388 nm—water solvation shell OH-(H_2_O)_1,4_ [51] (1374 nm)—-OH group of Ca(OH)_2_ [90] (1373–1375 nm)—-OH of portlandite phase; this band is useful for diagnosis of the initiation of hydration process [90] (1379.31 nm, 1st overt. of 3625 cm^−1^)—proton oscillation, H_2_O symmetric stretch in H+·(H_2_O)_6_ [83]
1385	(1383.13 nm 1st overt. of 3615 cm^−1^)—H_2_O symmetric stretch in H+·(H_2_O)_5_ [83] (1383.13 nm, 1st overt. of 3615 cm^−1^) Interwater/Double donor stretch (OH- (H_2_O)_4_) [88] (1385.12 nm, 1st overt. of 3609.8 cm^−1^) H_2_O symmetric stretch in proton hydrate H+(H_2_O)_4_ [91] (1385.50 nm, 1st over. of 3608.8 cm^−1^) H_2_O symmetric stretch in proton hydrate H+(H_2_O)_4_ [92] WAMACS C4: 1380–1388 nm—water solvation shell OH-(H_2_O)_1,4_ [51]
1391	(1391.21 nm 1st overt. of 3594 cm^−1^) H_2_O symmetric stretch in proton hydrate H+(H_2_O)_4_ [91,92] or trapped water 1396–1403 nm [69]
1397	(1396.6 nm, 1st overt. of 3580 cm^−1^) proton hydrates [H+·(H_2_O)_3_]—H_3_O+ free-OH stretch, 1st overt. [83] (1397 nm (7158 cm^−1^))—1st overtone of the free OH group trapped in the hydrophobic interior [93] WAMACS C5: water confined in the local field of ions 1396–1403 nm [52,69] (1397.23 nm (7157 cm^−1^))—interlayer OH- (stacked between sheets of Mg(OH)_2_) [78]
1410	1st overt. band of the OH stretching mode of free OH monomer [94] (1410.6 nm)—water species with no hydrogen bonds S_0_ [95] WAMACS C5: 1398–1418 nm—free water molecules S_0_
1428	(1428.6 nm) isolated H_3_O+ -OH stretch vibration, 1st overt. [96] 1st overtone of the fundamental OH stretching vibration in water; the water molecules are condensed in one or more layers on sorption sites in the amorphous region; related to stiffness and strength [77]
1441	WAMACS C7: 1432–1444 nm—water molecules with 1 hydrogen bond S_1_
1447	(1447 nm (6910 cm^−1^))—1st overt. of O−H stretching of the water OH hydrated to other water molecules (bulk state) [97] (1447.18 nm (6910 cm^−1^))—OH group involved in the OH⋯OH hydrogen bonding [98] (1447.18 nm, 1st overt. of 3445 cm^−1^)—stretching modes of surface H_2_O molecules or to an envelope of hydrogen-bonded surface OH groups [99] (1450.11 nm, 1st overt. of 3448 cm^−1^)—OH stretching vibrations of the water lattice in the hydrated calcium silicates and aluminosilicates (C–S–H and C–A–S–H) [90] WAMACS C8: 1448–1454 nm—solvation shell OH-(H_2_O)_4,5_
1460	WAMACS C9: 1458–1468 nm—water molecules with 2 hydrogen bonds S_2_
1472	(1470 nm)—chemically bound water in the hydrated calcium silicate phases [90] WAMACS C10: 1472–1482 nm—water molecules with 3 hydrogen bonds S_3_
1490	WAMACS C11: 1482–1495 nm—water molecules with 4 hydrogen bonds S_4_
1503	(1503.3 nm 1st overt. of 3326 cm^−1^)—OH stretching vibrations of hydrogen bonded water molecules participating in the crystal structure [100] (1503.3 nm 1st overt. of 3326 cm^−1^)—OH stretching vibration in Ice III [101] (1503.3 nm 1st overt. of 3326 cm^−1^)—strong intermolecular hydrogen bond [102] (1503.3 nm 1st overt. of 3326 cm^−1^)—water stretching vibrations in minerals, in connection with hydrogen defects (incorporation of hydrogen (protonation)) [103,104,105,106,107,108]
1515	WAMACS C12: 1506–1516 nm—combination of symmetric stretching and bending vibration ν1 + ν2, strongly bound water [51]
1534	(1534.21 nm, 1st overt. of 3259 cm^−1^)—hydrogen bonded hydroxyl groups (–O–H^δ+^⋯O^δ−^–) [109] (1534.21 nm, 1st overt. of 3259 cm^−1^)—the H–O stretching vibrations of the absorbent water [110] (1534.21 nm (6518 cm^−1^))—1st over. of hydrogen bonded O–H stretching [111] (1534.21 nm, 1st overt. of 3259 cm^−1^)—one of the 3 water stretching bands observed in carbonate mineral huanghoite by Raman spectroscopy (the other two being 1435 nm (3484 cm^−1^) and 1393 nm (3589 cm^−1^)) [103] (1534.21 nm, 1st overt. of 3259 cm^−1^)—sesquihydrate crystallite [112] (hydrate whose solid contains 3 molecules of water of crystallization per two molecules) (1534 nm)—one of 3 wavelengths used in multiple linear regression for predicting bread loaf volume (1506, 1534 and 1618 nm); measurement of some parameter related to volume independent of protein [113]
1559	(1557 nm) ionic bound water molecules 1st overt. [114] (1560 nm (3205 cm^−1^))—strongly hydrogen bonded water, water coordinated to cations [115] (1560 nm)—hydrogen bonded water [116] (1560 nm (6410 cm^−1^)) crystalline water ice feature [117]

**Table 6 molecules-27-07885-t006:** Summary of the water samples used in this study.

Water	Collection Site	Characteristics
W_dist_	—	High purity Wdist purified by ion exchange method and followed by distillation
W_tap_	Osaka City, Osaka Prefecture, Japan	Tap water collected mainly from the surface water of Lake Biwa and purified at a water treatment facility
W_shallow_	Water from the shallow underground source located at the dept of 40 m	Natural hot spring Yunosato, Hashimoto City, Wakayama Prefecture, Japan (https://www.spa-yunosato.com/yunosato_eng/ accessed on 1 November 2022)
W_mix_	Water that is a blend of two types of spring waters (90% water from the shallow source located at 50 m depth and 10% water from the deep source at 1187 m depth)	Natural hot spring Yunosato, Hashimoto City, Wakayama Prefecture, Japan (https://www.spa-yunosato.com/yunosato_eng/ accessed on 1 November 2022);

The mineral content of all four water types was determined by inductively coupled plasma mass spectroscopy (ICP-MS). The component analysis was conducted by the Metal team at Kyoto Municipal Institute of Industrial Technology and Culture (http://tc-kyoto.or.jp/about/organization/kinzoku/detail.html accessed on 14 November 2020).

**Table 7 molecules-27-07885-t007:** An example of the quality of Portland cement (chemical composition).

Chemical Name	Maximum Content Allowed (%)
MgO	5.0
SO_3_	3.5
Ignition Loss	5.0
Total alkali content	0.75

**Table 8 molecules-27-07885-t008:** An example of the chemical composition of standard sand.

Chemical Name	Content (%)
SiO_2_	98.4
Al_2_O_3_	0.4
Fe_2_O_3_	0.4
CaO	0.2
MgO	0.00
Na_2_O	0.01
K_2_O	0.01

**Table 9 molecules-27-07885-t009:** An example of the physical properties of standard sand.

Property	Value
Specific gravity in oven-dried condition	2.64
Sater absorption rate	0.42%
Unit volume mass	1.76 kg/L
Solid content	66.7%

## Data Availability

All data used in this study are available from the corresponding author on reasonable request.

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
