# Peer review of "Aquaphotomic Study of Effects of Different Mixing Waters on the Properties of Cement Mortar"

_molecules, 2022, doi:10.3390/molecules27227885_

Round 1
Reviewer 1 Report
The effect of water on the properties of cement mortar was studied by aquaphotomics studies. Water molecular network was studied using near infrared (NIR) spectroscopy, and the effect of different water on the drying shrinkage characteristics, as well as other characteristics, of the hardened cement was investigated. Chemometric methods including PCA and SIMCA were used for analyzing the water structures. The topic is interesting, and the manuscript was well prepared. Therefore, the paper can be accepted for publication after minor revisions.
1. More details can be included in the abstract, e.g., the technique (NIR spectroscopy) used and the conclusion obtained.
2. Studies of water structures using NIR spectroscopy has been well reported, particularly the efforts on the development of chemometric methods. Citation of these work is expected if possible.
Author Response
RESPONSES TO REVIEWERS FOR MANUSCRIPT TITLED:
" Aquaphotomic study of effects of different mixing waters on the properties of cement mortar"
Dear Editor,
Dear Reviewers,
Authors of the above-mentioned manuscript would like to thank you for the consideration and evaluation of the submitted study and for taking the time to read and review it.
Your corrections and comments were very useful and helped a lot to improve the manuscript. We are sending the revised and updated version of the manuscript along with this document. For easy follow-up of all the changes that were made, we are sending the corrections with highlighting the changes in yellow color. Authors deem that the content of the updated manuscript is accurate and hope that it will be accepted for publication.
Authors
----------------------------------------------------â‚°-----------------------------------------------------------
Reviewer #1: General remarks
The effect of water on the properties of cement mortar was studied by aquaphotomics studies. Water molecular network was studied using near infrared (NIR) spectroscopy, and the effect of different water on the drying shrinkage characteristics, as well as other characteristics, of the hardened cement was investigated. Chemometric methods including PCA and SIMCA were used for analyzing the water structures. The topic is interesting, and the manuscript was well prepared. Therefore, the paper can be accepted for publication after minor revisions.
Authors’ response
Dear Reviewer,
Thank you very much for the positive feedback about our manuscript. We are very grateful that you found the topic interesting, and that you think the manuscript was well prepared. We are also grateful for the constructive criticism and have revised the manuscript to improve the weak spots you indicated. We have changed the abstract and introduced new references, as instructed. Please find all the specific comments and our point-to-point replies below, where we describe performed revisions and where they can be found in the Manuscript. All the revisions in the Manuscript are highlighted in yellow for easy tracking. We hope that we successfully addressed all the issues and that paper can now be accepted for publication.
Thank you for investing your time and expertise to help us improve this Manuscript.
Authors
----------------------------------------------------â‚°-----------------------------------------------------------
Reviewer #1: Specific comments
|
Comment 1 |
More details can be included in the abstract, e.g., the technique (NIR spectroscopy) used and the conclusion obtained. |
|
Response 1 |
Thank you for this suggestion. We have enriched the abstract by adding details about the technique we used, methods of analysis and what were the main conclusions. Please find the revised abstract at L21-L37. |
|
Comment 2 |
Studies of water structures using NIR spectroscopy has been well reported, -particularly the efforts on the development of chemometric methods. Citation of these work is expected if possible. |
|
Response 2 |
Thank you for this suggestion. We have added several new references in the Introduction section, all of which are focused on development of preprocessing and chemometrics techniques for better understanding of the water molecular structures based on the NIR spectra. Please find the revision at L134-137. |
Authors are very grateful for the comments of Reviewer 1.

Reviewer 2 Report
In this study, author investigated the effect of four different types of water (two spring-, mineral waters, tap water and distilled water) on the drying shrinkage of the hardened cement by comparing the material properties of the concrete specimens and analyzing the molecular structure of the water and cement mortar using aquaphotomics.There are some issues needed to be clarified before it can be accepted for publication in molecules:
1. Please do not use the first person in scientific papers.
2. The research contributions of the paper should be articulated more clearly. The abstract is not representative of the content and contributions of the paper. The abstract does not seem to properly convey the rigor of research.
3. The manuscript should be very carefully checked to avoid any errors. The language should be checked throughout the text and any grammar mistakes should be corrected.
4. The authors should explain or annotate the abbreviations at their first use site. It is beneficial to readers who are not familiar with this field. E.g. w/c should be w/c ratio.
5. The authors explain the effect of the water-binder ratio on concrete in the introduction, which is correct, but a fuller mechanism and ref. should also be elaborated. The water-to-binder ratio also has a great influence on the growth of hydration products. E.g. A novel development of green UHPC containing waste concrete powder derived from construction and demolition waste. In addition, the effects of shrinkage deformation should also be addressed here. E.g. Improvement mechanism of water resistance and volume stability of magnesium oxychloride cement: A comparison study on the influences of various gypsum.
6. Figure 5. Please give the meaning of the horizontal and vertical coordinates.
7. Table 2. Pls use (%).
8. Table 5. The authors are asked to double-check these results and references, the reviewers found some of them to be erroneous. The position of the absorption band does not correspond to the functional group.
9. Please make sure your conclusions' section underscore the scientific value added of your paper, and/or the applicability of your findings/results, as indicated previously. Please revise your conclusion part into more details. Basically, you should enhance your contributions, limitations, underscore the scientific value added of your paper, and/or the applicability of your findings/results and future study in this session.
Author Response
RESPONSES TO REVIEWERS FOR MANUSCRIPT TITLED:
" Aquaphotomic study of effects of different mixing waters on the properties of cement mortar"
Dear Editor,
Dear Reviewers,
Authors of the above-mentioned manuscript would like to thank you for the consideration and evaluation of the submitted study and for taking the time to read and review it.
Your corrections and comments were very useful and helped a lot to improve the manuscript. We are sending the revised and updated version of the manuscript along with this document. For easy follow-up of all the changes that were made, we are sending the corrections with highlighting the changes in yellow color. Authors deem that the content of the updated manuscript is accurate and hope that it will be accepted for publication.
Authors
----------------------------------------------------â‚°-----------------------------------------------------------
Reviewer #2: General remarks:
In this study, author investigated the effect of four different types of water (two spring-, mineral waters, tap water and distilled water) on the drying shrinkage of the hardened cement by comparing the material properties of the concrete specimens and analyzing the molecular structure of the water and cement mortar using aquaphotomics. There are some issues needed to be clarified before it can be accepted for publication in molecules.
Authors’ response:
Dear Reviewer,
We are very grateful for your constructive criticism and the opportunity to revise the manuscript. You provided some general and specific comments about needed revisions. We have addressed all these issues (as will be explained in the answers to the specific comments) and revised the paper according to your instructions. All the changes in the revised manuscript are highlighted in yellow, and in responses to specific comments we explained how we changed the Manuscript, and at which line number the revisions can be found. Please find the responses to specific comments below this Letter. We hope that we successfully corrected all the issues and that paper can now be accepted for publication.
Thank you for investing Your time and expertise to help us improve this Manuscript.
Authors
----------------------------------------------------â‚°-----------------------------------------------------------
Specific comments:
|
Comment 1 |
Please do not use the first person in scientific papers. |
|
Response 1 |
Thank you for this comment. Yes, we are very well aware of the rules of scientific writings, we regret it slipped our attention in this paper. Thank you for pointing this out. We have corrected all instances of using first person in the revised manuscript. Please find such revisions at L128, L143, L146, L550, L558, L563, L627-628, L641, L651, L668 and L848. |
|
Comment 2 |
The research contributions of the paper should be articulated more clearly. The abstract is not representative of the content and contributions of the paper. The abstract does not seem to properly convey the rigor of research. |
|
Response 2 |
Thank you for requesting this information. We have revised the abstract to better show which measurement methods and data analysis techniques were used. And we formulated better the contributions and conclusions of the paper. Please find the revised abstract at L21-L37. |
|
Comment 3 |
The manuscript should be very carefully checked to avoid any errors. The language should be checked throughout the text and any grammar mistakes should be corrected. |
|
Response 3 |
Thank you for this advice. We have checked the manuscript thoroughly for language errors. Since there were many, we will not list the revisions here, but all the revisions performed in response to this particular request are highlighted in green. |
|
Comment 4 |
The authors should explain or annotate the abbreviations at their first use site. It is beneficial to readers who are not familiar with this field. E.g. w/c should be w/c ratio. |
|
Response 4 |
Thank you for this observation. We have defined all the abbreviations at first use site, please see revisions at L46, L158, L160 and L162. |
|
Comment 5 |
The authors explain the effect of the water-binder ratio on concrete in the introduction, which is correct, but a fuller mechanism and ref. should also be elaborated. The water-to-binder ratio also has a great influence on the growth of hydration products. E.g. A novel development of green UHPC containing waste concrete powder derived from construction and demolition waste. In addition, the effects of shrinkage deformation should also be addressed here. E.g. Improvement mechanism of water resistance and volume stability of magnesium oxychloride cement: A comparison study on the influences of various gypsum. |
|
Response 5 |
Thank you for this suggestion! The papers you mentioned are really useful for us! We have provided the revision in the Introduction section, where we gave more explanation on the water-binder ratio and the influence on the growth of hydration products, and we also provided some new references, including the two suggested by the reviewer. Please find the revision at L52-L67. |
|
Comment 6 |
Figure 5. Please give the meaning of the horizontal and vertical coordinates. |
|
Response 6 |
We have changed the Figure 5 to include the percentage of explained variance by first two principal components PC1 and PC2, which define horizontal and vertical axes of each scores plot, as is the usual way when reporting the results of PCA analysis. We provided more explanation about the meaning of principal components in the figure legend by explaining that these two axes are defined by the combination of original variables, orthogonal to each other, that retain more than 95% of the information in the original dataset. The meaning of PC1 can be related to changes in water in cement over the course of days, while in the case of PC2 to changes in water during first 24 hours. Please find the revision at L287-L293. We have also added additional explanation in the text at L279-280 before we refer to the Figure 5. |
|
Comment 7 |
Table 2. Pls use (%). |
|
Response 7 |
Thank you, we have corrected the Table 2, to display percentages as (%). Please find the revision at L295. We also corrected similar mistakes in Table3 (please see L416) and Table 4 (please, see L445). |
|
Comment 8 |
Table 5. The authors are asked to double-check these results and references, the reviewers found some of them to be erroneous. The position of the absorption band does not correspond to the functional group. |
|
Response 8 |
Thank you for noticing this! We have double checked the references, and corrected the mistakes, some of the incorrect references had to be removed. We also provided more precise values for the positions of absorbance bands after they were recalculated from wavenumbers and/or fundamental bands. Please find the revisions marked in yellow in Table 5 (starting at L479). |
|
Comment 9 |
Please make sure your conclusions' section underscore the scientific value added of your paper, and/or the applicability of your findings/results, as indicated previously. Please revise your conclusion part into more details. Basically, you should enhance your contributions, limitations, underscore the scientific value added of your paper, and/or the applicability of your findings/results and future study in this session. |
|
Response 9 |
Thank you for this remark. We have revised the conclusion as advised and expressed the main conclusions in a numbered list. We have also improved the description of scientific value, and the limitations of the study and how they can be overcome in future. Please see the revision at L864-L939, L941-944, L946, and L953-962. |
Authors are grateful for the comments of Reviewer 2.

Round 2
Reviewer 2 Report
It can be accepted.